# Structural insights into Frizzled3 through nanobody modulators

James Hillier[1,4], Yuguang Zhao [1,4] ✉, Loic Carrique [1], Tomas Malinauskas [1], Reinis R. Ruza[1], Tao-Hsin Chang [1], Gangshun Yi [1], Helen M. E. Duyvesteyn[1], Jing Yu[1], Weixian Lu[1], Els Pardon [2,3], Jan Steyaert [2,3], Yanan Zhu [1], Tao Ni [1] & E. Yvonne Jones [1] ✉

The Wnt receptor Frizzled3 (FZD3) is important for brain axonal development and cancer progression. We report structures of FZD3 in complex with extracellular and intracellular binding nanobodies (Nb). The crystal structure of Nb8 in complex with the FZD3 cysteine-rich domain (CRD) reveals that the nanobody binds at the base of the lipid-binding groove and can compete with Wnt5a. Nb8 fused with the Dickkopf-1 C-terminal domain behaves as a FZD3-specific Wnt surrogate, activating β-catenin signalling. The cryo-EM structure of FZD3 in complex with Nb9 reveals partially resolved density for the CRD, which exhibits positional flexibility, and a transmembrane conformation that resembles active GPCRs. Nb9 binds to the cytoplasmic region of FZD3 at the putative Dishevelled (DVL) or G protein-binding site, competes with DVL binding, and inhibits GαS coupling. In combination, our FZD3 structures with nanobody modulators map extracellular and intracellular interaction surfaces of functional, and potentially therapeutic, relevance.

Wnt signalling pathways are fundamental for embryonic development, adult tissue homeostasis and regeneration[1]. Secreted Wnt ligands bind to the extracellular domains of FZD receptors and low-density lipo-protein receptor-related protein 5/6 (LRP5/6) co-receptors, leading to receptor clustering and β-catenin-dependent (canonical) signalling[2,3]. Alternatively, Wnts may engage FZDs with other receptors, such as receptor tyrosine kinase-like orphan receptor 2 (ROR2) or receptor tyrosine kinase (RYK), to initiate alternative signalling pathways that are β-catenin independent (non-canonical)[4].

FZD3, together with the closely related FZD6, forms a distinct subset within the Frizzled family, which has been reported to signal exclusively through non-canonical Wnt pathways, without eliciting β-catenin-dependent signaling[5]. Understanding how different Wnts recognize specific FZDs to initiate distinct signalling pathways is a central question in Wnt research. The N-terminal extracellular CRD of FZD is the primary Wnt-binding site and determinant of specificity[6], but the extracellular linker region has also been reported to contribute[7].

The structures of several FZD CRDs have been determined[8–11]. However, information has been lacking on the detailed architecture and binding characteristics of the FZD3/6 subset. Structural analyses of the FZD transmembrane domain (TMD) have been reported for FZD4, FZD5 and FZD7, revealing the GPCR-like fold comprised of seven transmembrane helices[12–14]. Full length receptor structures including the extracellular region (CRD and linker) of FZD5 and FZD7 lack interpretable density for the CRD presumably because of a substantial degree of flexibility in its position relative to the TMD[12,13]. Consistent with this conclusion, a cryo-EM-based low resolution analysis of the xWnt8-FZD5 complex showed extreme flexibility between the Wnt-bound CRD and the TMD[12]. The FZD3/6 subset has relatively short linker sequences between the CRD and TMD and is therefore potentially a more favourable system in which to visualise the CRD and linker regions of a full-length FZD to advance our understanding of Wnt-FZD interactions, specificity determination, and the mechanisms underpinning receptor activation.

[1]Division of Structural Biology, Wellcome Centre for Human Genetics, University of Oxford, Oxford, UK. [2]Structural Biology Brussels, Vrije Universiteit Brussel, VUB, Brussels, Belgium. [3]VIB-VUB Centre for Structural Biology, VIB, Brussels, Belgium. [4]These authors contributed equally: James Hillier, Yuguang Zhao. ✉e-mail: yuguang@strubi.ox.ac.uk; yvonne@strubi.ox.ac.uk

FZD receptors transmit signals to downstream effectors through DVL in both canonical and non-canonical pathways[15]. DVL has three domains: an N-terminal Dishevelled and Axin (DIX) domain; a central Postsynaptic density protein-95, Disk large tumour suppressor, Zonula occludens-1 (PDZ) domain; and a C-terminal Dishevelled, Egl-10, and Pleckstrin (DEP) domain. The DEP domain is responsible for FZD binding[16], however, the DEP-FZD interaction has not been characterised structurally.

A number of observations point to FZD3 as a therapeutic target. Functionally, FZD3 is crucial for axon growth and guidance in the nervous system[17,18] and in migration of neural crest cells[19]. FZD3 also plays a significant role in establishing planar cell polarity (PCP) in the vertebrate neural plate[20], while FZD3 knock-out mice exhibit a loss of fibre tracts in the rostral central nervous system[21] or thinning of spinal motor nerves[18]. Dysregulation of FZD3 may be involved in neurodegenerative Parkinson's disease[22] and Hirschsprung disease[23]. FZD3 is upregulated in many types of cancer, such as hepatocellular carcinoma[24] and B-cell chronic lymphocytic leukemia[25]. In addition to its role in melanocyte and hair follicle development[26,27], FZD3 is a critical regulator of human melanoma tumorigenesis. Knocking down FZD3 suppresses the growth and metastasis of human melanoma[28].

Wnt receptor antibodies and other synthetic Wnt receptor-binding proteins are being developed with the intention of suppressing Wnt signalling by competing with Wnt for receptor binding, with the potential for tumour supresion[29–31]. Alternatively, synthetic Wnt agonists could potentially be used to stimulate stem cell activity for the treatment of various degenerative conditions[32]. Although not all protein ligands (such as cache domain containing 1, CACHD1) that link FZD and LRP5/6 could trigger Wnt/β-catenin signalling[33], it has been shown many chimeric proteins (especially bispecific antibodies) that simultaneously bind both FZD and LRP5/6 can serve as a Wnt mimic to activate signalling[34–36]. These Wnt surrogates may possess advantages over native lipophilic Wnt ligands by exhibiting greater FZD specificity and biochemical tractability.

To develop FZD3-specific modulators and build a structural framework for potential therapeutic interventions, here we show a panel of FZD3 specific nanobodies and the structures of the FZD3 CRD and full-length FZD3 in complex with two high-affinity nanobodies. Our structures, combined with biophysical and cellular assays, illuminate the mechanisms of FZD3 signalling and how it could be modulated by nanobodies.

## Results

### Nanobodies against FZD3

We expressed and purified full-length FZD3 and used it as an antigen to immunize llamas. Seven camelid antibodies (nanobodies, Nb1, 3–6; 8–9) were identified. Their amino acid sequences are shown in Supplementary Fig. 1. Nb1 and Nb3–6 could be grouped together based on similar complementarity-determining region 3 (CDR3) sequences, suggesting that they may recognize the same region of FZD3. In contrast, Nb8 and Nb9 exhibit distinct CDR3 sequences.

We first tested the nanobodies for binding to the FZD3 CRD and found that only Nb8 binds to this domain. To further test the binding specificity, we produced biotinylated samples of all 10 human Frizzled CRDs to test their binding to Nb8 using Biolayer Interferometry (BLI). Apart from FZD3 CRD, only FZD6 CRD showed weak binding to Nb8. All other FZD CRDs did not bind to Nb8 (Supplementary Fig. 2). We next sought to quantify the affinity of Nb8 for the FZD3/6 CRDs. Nb8 binds to FZD3 CRD with high affinity, Kd $19.2 \pm 1.2$ nM (Fig. 1a). It binds to FZD6 CRD with much lower affinity, Kd $1200 \pm 200$ nM (Fig. 1b). We showed FZD3 CRD binding to human Wnt5a (Kd $83 \pm 4$ nM, Fig. 1c) and then tested whether Nb8 and Wnt5a compete for FZD3 CRD binding. A similar amount of biotinylated FZD3 CRD was immobilized onto two BLI sensor tips. Then one tip was dipped into Nb8, while the other was dipped into a bovine serum albumin (BSA) solution as a control. After a

dissociation step, both tips were further dipped into Wnt5a. While the BSA-dipped control tip showed prominent binding to Wnt5a, the Nb8-dipped tip showed reduced binding to Wnt5a, indicating that Nb8 and Wnt5a compete for FZD3 binding (Fig. 1d).

As the other nanobodies did not bind to the FZD3 CRD, we then produced biotinylated full-length FZD3 to assess for binding using BLI. Nb1 (representative of the Nb1 and Nb3−6 group) and Nb9 were tested. Nb9 bound with high affinity (Kd $32 \pm 2.8$ nM, Fig. 1e). Nb1 also bound full length FZD3, but the response was weaker compared to Nb9 (Supplementary Fig. 3). To assess the specificity of Nb9 binding to FZD3, we measured binding to FZD6 and found a much lower affinity (Kd $410 \pm 41$ nM, Fig. 1f). As Nb8 and Nb9 bound to FZD3 with the highest affinity, we used these two nanobodies for further structural and functional studies of FZD3.

### Crystal structure of FZD3 CRD in complex with Nb8

FZD3 CRD did not crystallize in isolation. However, it crystallized readily in the presence of Nb8. We determined the crystal structure of the FZD3 CRD in complex with Nb8 to 1.8 Å resolution (Supplementary Table 1). Although the FZD3/6 subgroup CRD sequences have diverged from other Frizzled family members, the structure revealed an evolutionarily conserved FZD CRD fold. FZD3 CRD could be superimposed with CRDs from FZDs 2, 4, 5, 7, 8 (PDB: 6C0B[9], 5BPQ[8], 5URY[10], 5T44[10], 6TFM[11]) with a root-mean-square deviation (RMSD) of 1.4, 1.6, 1.5, 1.2, 1.4 Å for 113 Cα aligned atoms, respectively (Supplementary Fig. 4a, b).

The FZD3 CRD and Nb8 interaction interface is well resolved in the structure (Supplementary Fig. 4c). Consistent with their high binding affinity, the CRD and Nb8 respectively contribute 1300 and 1360 Å² of their surface area to the interaction interface in the complex. The interactions are predominantly hydrophobic (Supplementary Fig. 5). For example, FZD3 M116 forms hydrophobic interactions with Nb8 Y37, N96 and L98. Hydrogen bonds also form between FZD3 K112 and Nb8 E103 and D105. FZD CRDs display a lipid-binding groove, which serves as the binding site for the palmitoleate moiety of Wnt proteins[6]. The complex structure shows how Nb8 'wraps around' the FZD3 CRD at the base of the lipid-binding groove using a concave surface formed by the three CDRs as well as their surrounding sequences. The binding epitope is moderately to poorly conserved among the 10 human FZD CRDs (Fig. 2a, b and Supplementary Figs. 4, 5), providing a rationale to explain the high degree of FZD3-binding specificity exhibited by Nb8. In particular, residue M116 (unique to FZD3) makes multiple interactions with Nb8 residues, and K112 (present only in the FZD3/6 subfamily) contributes electrostatic complementarity (Supplementary Fig. 5).

Superposition of our FZD3 CRD-Nb8 complex with the FZD8 CRD-Wnt3 assembly (PDB 6AHY)[37] suggests that Nb8 (specifically, T52) would impede access of the Wnt palmitoleic moiety to the FZD3 CRD lipid-binding groove (Fig. 2b, c). Additionally, the groove in the FZD3 CRD is narrower than that observed in the FZD8 Wnt3 complex (Fig. 2c). The end of helix 3 in the FZD3 CDR is shifted closer to helix 1 by some 5.9 Å (as measured for the Cα atoms of F117 in FZD3 and Y125 in FZD8) compared to its position for a lipid-accommodating groove in FZD8 (Fig. 2c). It is possible this groove narrowing conformation exists when no Wnt ligand engaged, but also possible it might be due to the nanobody binding, given the extensive interactions between Nb8 (mediated by Y37 from CDR1; L47 and T52 from CDR2, and N96 from CDR3) with the end of the FZD3 CRD helix 3. This combination of predicted steric hindrance effects, both direct and indirect (through groove narrowing), is consistent with our binding data showing that Nb8 competes with Wnt5a for FZD3 CRD binding (Fig. 1d). We also analysed the potential compatibility of Nb8 binding with CRD dimerization as observed in the structure of the FZD8 CRD−Wnt3 complex[37]. Our structural superposition indicates that Nb8 binding would hinder formation of the CRD dimerization interface (Fig. 2d). Our FZD3 CRD alone appears to be monomeric as judged from the gel filtration profile (Supplementary Fig.4d). However, this does not exclude the possibility of it forming a

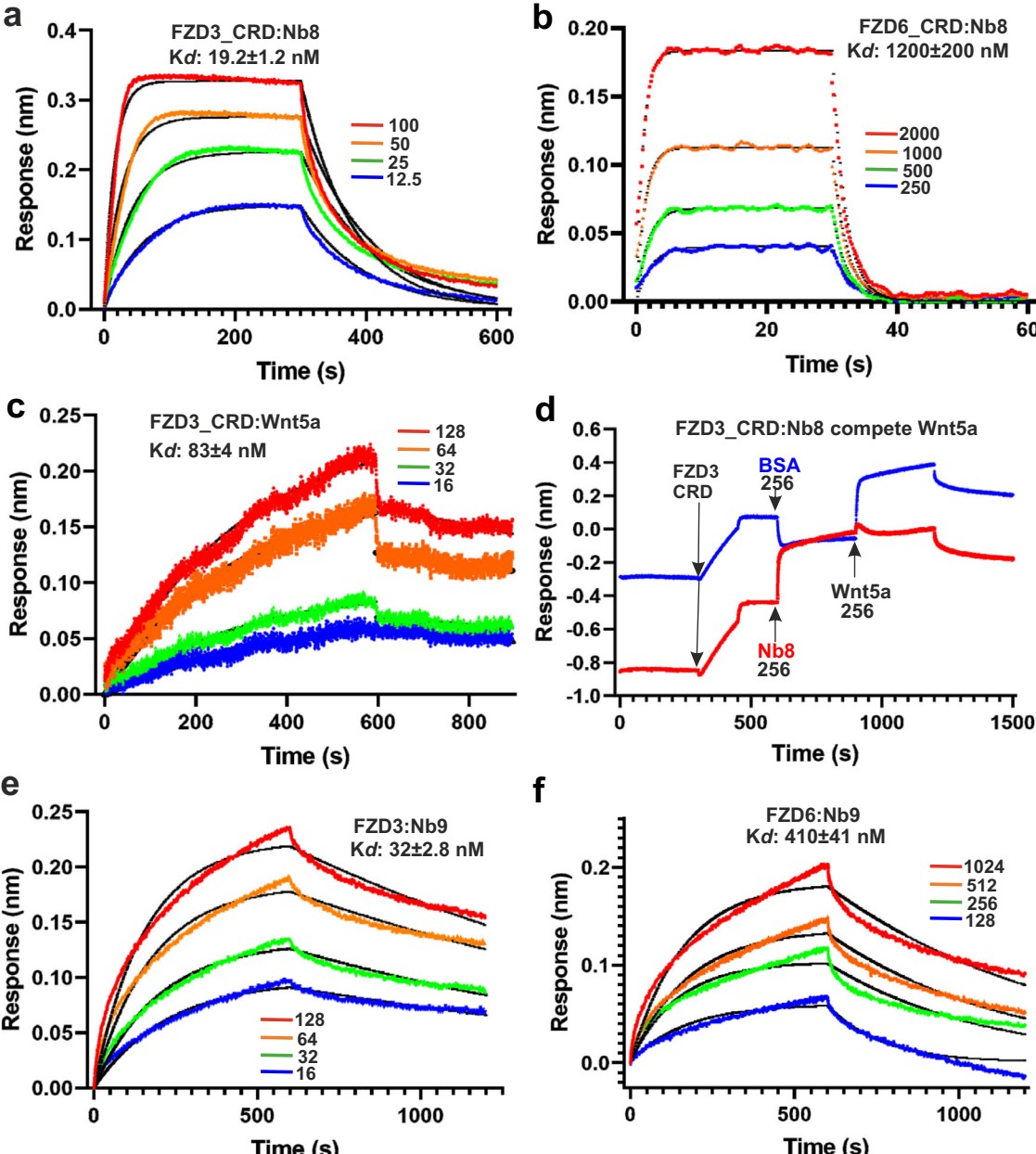

**Fig. 1 | BLI sensorgrams for nanobodies or Wnt5a binding to FZD3/6. a** Nb8 binding to immobilized FZD3 CRD. Different concentrations are shown by coloured lines with indicated concentrations (nM). Fitted curves are shown by black lines. **b** Nb8 binding to immobilized FZD6 CRD. **c** Wnt5a binding to immobilized FZD3 CRD. **d** Competition assay for Nb8 and Wnt5a. The tips were loaded with FZD3 CRD, dipped in 256 nM Nb8 or BSA control, then dipped in 256 nM of Wnt5a. Nb9 binding to immobilized full length FZD3 (**e**) and FZD6 (**f**).

dimer upon ligand engagement, similar to the behaviour of FZD4 CRD, which itself is monomeric[8], but forms a dimer in response to Wnt5a binding[38]. Other FZD CRDs dimerization induced by Wnt palmitoleic lipid have also been documented[10,37], and its importance in Wnt signalling further demonstrated by multivalent Wnt surrogates[39]. Taken together, Nb8 might inhibit both FZD3-Wnt, and FZD3-FZD3 interactions.

## A Nb8 derived Wnt surrogate initiates β-catenin signalling

Having generated a highly specific FZD3 CRD nanobody, we sought to determine whether it could be engineered to form a Wnt surrogate. We fused Nb8 with the Dickkopf1 C-terminal module (DKK1C, which binds to LRP6 propeller domain 3) to facilitate recruitment of LRP6, a strategy previously used for other synthetic FZD agonists[34] (shown

schematically in Fig. 2e). We added a C-terminal Fc domain to dimerise this bivalent unit. The functionality of the dimeric fusion protein (Nb8_DKK1C) was then investigated using the well-established TOP-Flash luciferase assay[40] to measure β-catenin dependent signalling activity. When Nb8_DKK1C was transfected with a control plasmid (pcDNA3) into HEK293T cells lacking any FZD (FZD1–10[-/-]), no signalling was detected. However, when co-transfected with wild type FZD3, Nb8_DKK1C induced luciferase activity over 40-fold above basal levels (Fig. 2f). This is intriguing, as FZD3 is generally considered to be a non-canonical Wnt receptor, and a previous study was unable to identify any Wnts that could stimulate β-catenin signalling via FZD3[5]. However, our data are consistent with the recent development of a FZD6 targeting Wnt surrogate that can robustly activate β-catenin signalling[36]. The signalling level by our Nb8_DKK1C surrogate is comparable to that

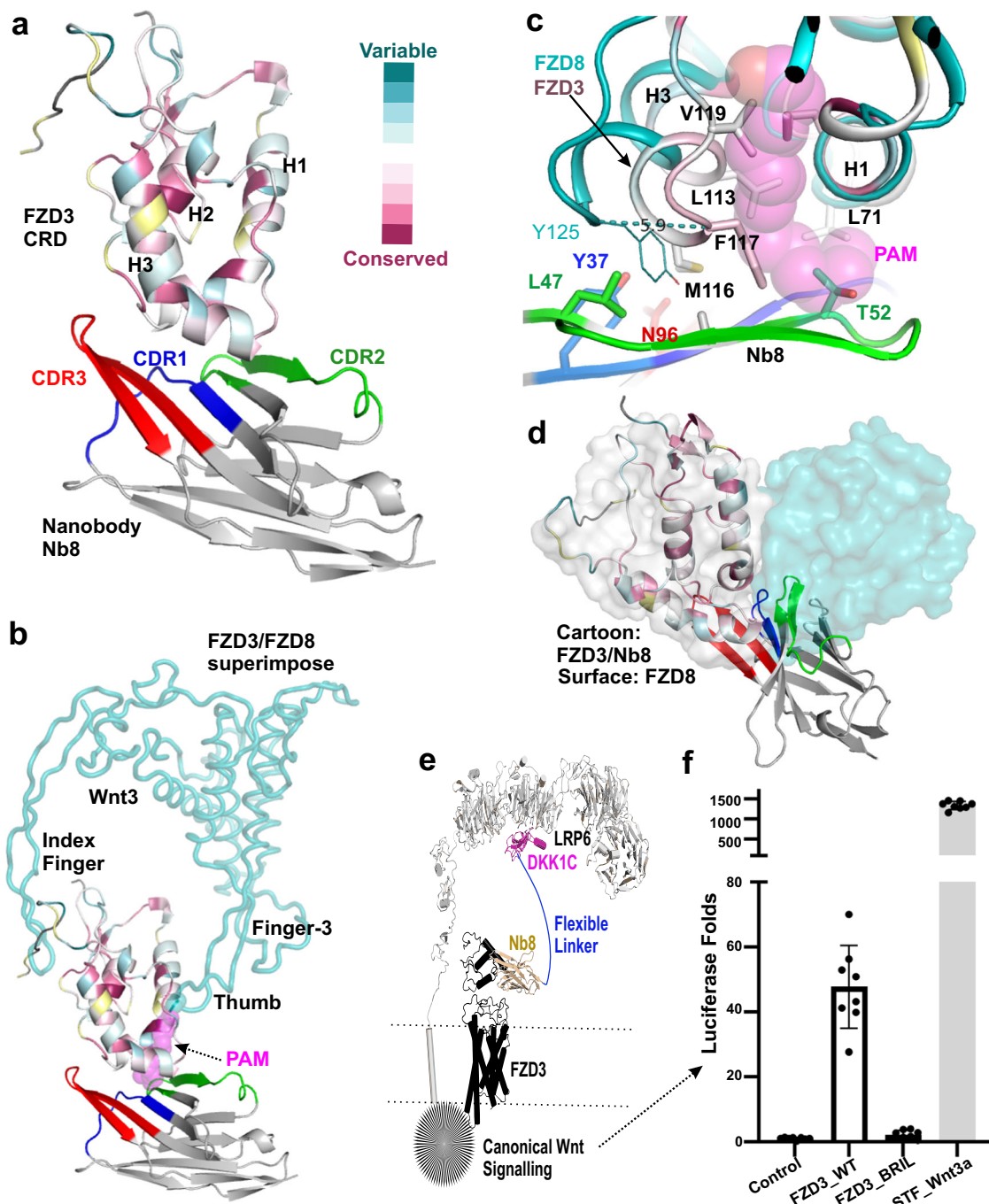

**Fig. 2 | The crystal structure of the FZD3 CRD in complex with Nb8 and Wnt surrogate luciferase assay. a** Cartoon representation of FZD3 CRD, coloured according to consurf alignment of the 10 human FZD CRD sequences. Nb8 coloured in blue (CDR1), green (CDR2) and red (CDR3). **b** FZD3 CRD superimposed with FZD8-Wnt3 complex (PDB 6AHY). Wnt3 is shown as teal tube and the lipid (palmitoleic acid, PAM) as magenta spheres. The aligned FZD8 CRD is hidden for clarity. **c** Close up view of FZD3 lipid binding groove compared to the groove of FZD8 bound to Wnt3. **d** FZD3 CRD superimposed with FZD8 dimer (as light grey and teal surface, PDB 6AHY). **e** A schematic representation of the Wnt surrogate encompassing comprising Nb8 (in wheat), which binds to FZD3 (black) linked (via a flexible linker in blue) to the C-terminal module of Dickkopf1 (DKK1C, magenta) which binds to the Wnt co-receptor LRP6/5 (grey). **f** Luciferase reporter-based Wnt signalling assays in FZD1–10-/- HEK293T cells and STF control. Normalized TOP-Flash luciferase activity was induced by the Wnt surrogate. Cells were transfected with either wild-type FZD3 or FZD3_Bril (denotes FZD3 with a Bril fusion in its ICL3 between TM-5 and TM-6). The STF cells were induced with L-cell Wnt-3a conditional media (bar with shading) as a positive control for comparison. Each dot indicates a sample value. ($n$ = 9, three biological experiments with triplicates, the error bars are presented as mean values +/-SD).

of antibody-based FZD6-targeting Wnt surrogates[36] where it is sufficient to be functional. However, the overall measured TOP FLASH level by our Nb8_DKK1C surrogate is far lower compared to the level induced by Wnt3a-containing conditioned media in STF (SuperTopFlash)[41] cells containing multiple FZDs (Fig. 2f). These results suggest that members of the 'non-canonical' FZD3/6 subset, similarly to 'canonical' FZDs, can transduce β-catenin signalling when suitably oligomerised with LRP5/6. To further confirm the involvement of the FZD3 TMD and intracellular region in transducing the Wnt signal, we engineered a FZD3_BRIL (the thermostabilised apocytochrome

b562 from *Escherichia coli*) fusion construct, in which BRIL is inserted into intracellular loop 3 (ICL3). A similar FZD5 construct was previously demonstrated to adopt an inactive TMD conformation[12]. When FZD3_BRIL was tested with the Nb8_DKK1C surrogate, no TOP-Flash luciferase activity was detected (Fig. 2f), confirming that the wild type intracellular domain of FZD3 and, specifically, ICL3 are essential for the β-catenin signalling.

## Cryo-EM structure of FZD3 in complex with Nb9 megabody

We sought to determine the full length structure of FZD3 and to characterize the basis of its interaction with Nb9. We engineered Nb9 into a megabody[42] by fusing Nb9 with the bacterial α-galactosidase (YgjK scaffold) to facilitate a high resolution single-particle cryo-EM analysis. The scaffold is fused on the opposite side of the nanobody to its CDRs, thus minimizing the possibility of affecting antigen binding. For a similarly fused megabody (with a nanobody targeting Hedgehog acyltransferase) the fusion has been demonstrated to have little effect on binding affinity[43]. This scaffold increased the size of the FZD3-containing complex by an additional ~100 kDa, enabling us to determine the FZD3-Nb9 megabody structure at a global resolution of 2.9 Å (Fig. 3a and Supplementary Fig. 6).

## The FZD3 extracellular domain and Wnt specificity

Cryo-EM density for the FZD3 TMD (Fig. 3a and Supplementary Fig. 7) and Nb9 megabody allowed us to unambiguously model these protein regions. Although the weak extracellular density (even after the improvement with 3DFlex reconstruction[44]) is not sufficient for confidently modelling the FZD3 CRD structure, the presence of CRD in this region is a certainty. Since Wnt and the Nb8 (raised against full length FZD3 antigen) should bind without sterically clashing with the TMD, we proposed a CRD orientation (Fig. 3b), whereby our crystal structure can be most likely placed into the density.

For this CRD orientation, interestingly, the modelled Wnt5a (Wnt3-FZD8, PDB code 6AHY, used as template) shows a hairpin loop, referred to as Wnt finger-3 (Fig. 2b, Supplementary Fig.8f), proximal to the FZD linker domain (Fig. 3c). Wnt finger-3 is a distinctive feature involved in Wntless (WLS) binding[45,46] but its role, if any, in FZD interaction is unknown. On the other hand, the FZD linker domain has been demonstrated to be important for Wnt ligand recognition[7]. Specifically, replacement of the linker domain of FZD4 with that of FZD3, reduced FZD4-Wnt3a affinity and canonical signalling. Conversely, the replacement of the FZD6 linker domain with that from FZD1 resulted in the loss of Wnt5a binding and non-canonical signalling[7]. How differences in the FZD linker domain sequence, and thus structure, relate to Wnt selectivity is not understood. Our structural observations suggest that the Wnt finger-3 hairpin may contribute to the recognition of the FZD linker domain. To gain experimental evidence, we synthesized the biotinylated Wnt5a finger-3 peptide and FZD3 linker peptide (Supplementary Fig. 8a, d). The BLI data shows dose dependent responses of interaction with affinity Kd: $31.0 \pm 4.1$ μM (Fig. 3f), while the BSA control shows no or negligible responses with the same concentrations (Fig. 3g). This validates our proposed model for the FZD3 CRD orientation. It shows a Wnt finger-3 interacting with a FZD through a linker domain and sheds light on how FZD linker domain may contribute to Wnt signalling selection. Two prominent hairpin loops of Wnt bind to the FZD CRD[6], namely the "thumb" (from which the palmitoleate protrudes) and the "index finger". Sequence alignments show relatively high levels of conservation between Wnts for the thumb and index finger regions. In contrast, there are substantial differences in the finger-3 sequences, for example between Wnt3a and Wnt5a (Supplementary Fig. 8). We further explored interactions between Wnt5a finger-3 and FZD3 using AlphaFold. The AlphaFold model suggests multiple possible interactions between the two proteins, including residues unique to the FZD3 linker, L169 and I171, with the Wnt5a finger-3 residues A167 and L173

(Supplementary Fig. 8f). Thus, both our binding data and AlphaFold modelling suggest Wnt finger-3 may contribute to determining Wnt-FZD specificity.

## The conformational state of the Nb9 bound FZD3 TMD

The FZD3 TMD was well resolved in cryo-EM maps (Fig. 3a, b, and Supplementary Fig. 7) revealing the canonical GPCR fold of 7 transmembrane helices (TM1-7). We do not observe any density for amphipathic helix 8, as seen in FZD7[13] or FZD5[12]. TMDs of GPCRs can exhibit active or inactive states, primarily based on movement of TM6. Superpositions of the FZD3 structure with representative active FZD7 (PDB 7EVW)[13] and inactive FZD5 (PDB 6WW2)[12] indicate that Nb9 binding stabilizes FZD3 in an active conformation (Fig. 3d). The lower portion of FZD3 TM6 appears to be moved outwards (Fig. 3d), in a similar way to that observed in G protein bound FZD7[13]. For example, at the cytoplasmic end of TM6, FZD3 K413$^{6.25}$ has a Cα distance of 6.4 Å to FZD5 K442$^{6.25}$ but only 1.6 Å to FZD7 K463$^{6.25}$, (superscript numbers refer to the Ballesteros and Weinstein numbering system), despite the overall structures being similar among the three (FZD3 shows an RMSD of 1.6 Å and 1.8 Å to active FZD7 and inactive FZD5 respectively, when superposed with all the Cα atoms of the 332 residues from TMD helices and loops). Given the conformational similarity between FZD3 and FZD7, our structure suggests that FZD3 may possess a similar activation mechanism to FZD7. The previously identified F class GPCR molecular switch[47] involving residues R420$^{6.32}$ and W497$^{7.55}$ is conserved in FZD3, with the π-cation interaction maintained (Supplementary Fig. 9). However, the polar interactions in our FZD3 structure are absent as in the FZD7 in complex with G proteins (distance of 6.2 Å between R$^{6.32}$ and the backbone oxygen atom of the W$^{7.55}$, while in the inactive FZD5 the distance is 3.8 Å, Supplementary Fig. 9b). The kink residue P431$^{6.43}$, conserved in all FZDs but not in SMO, is believed to contribute to the outward movement of the lower part of TM6[13,47]. Finally, Y428$^{6.40}$ forms π–π interactions with W304$^{3.43}$ to further stabilize the active conformation of TM6 (Supplementary Fig. 9a).

Our FZD3 structure reveals a prominent pocket in the TMD that could potentially accommodate a small molecule (Fig. 3e). The development of small molecules targeting FZDs is lagging behind other GPCRs. However, the small molecule inhibitor F7H has recently been identified based on in silico docking to the FZD7 structure[48]. Multiple residues proposed to mediate interactions between F7H and FZD7 are conserved in FZD3 and contribute to the surface of the FZD3 TMD pocket: Y245$^{2.51}$, V248$^{2.54}$, M288$^{3.27}$, Y294$^{3.33}$, L379$^{5.51}$, Y439$^{6.51}$, and K483$^{7.41}$ (Fig. 3e). This comparison suggests that similar to FZD7, FZD3 contains a potentially druggable TMD pocket.

## Nb9 binding to FZD3

Nb9 binds to the cytoplasmic region of FZD3 in a pocket formed by the intracellular parts of the TM helices and their intracellular loops (ICLs) (Fig. 4a). Nb9 uses all three CDRs to interact with the FZD3 pocket residues (Fig. 4a–c). The positively charged pocket allows some hydrogen bonds and electrostatic interactions, for example between FZD3 K318, S321 and K413, and Nb9 N29, D31 and D102 (Supplementary Fig. 10). Hydrophobic interactions also make a major contribution to the binding interface between FZD3 and Nb9. For example, W100 from the tip of Nb9 CDR3 is deeply embedded in the FZD3 pocket, forming a π-π stacking interaction with W311$^{3.50}$, as well as hydrophobic interactions with F417$^{6.39}$, P234$^{2.40}$, and A315$^{3.54}$ of the FZD3 (Supplementary Fig. 10). Multiple FZD3 residues that mediate FZD3-Nb9 interactions are conserved across FZD family members (Supplementary Fig. 11). These include ICL1 residues R230, F231, P234, and E235; ICL2 residues K318, W319, E320, and A321; TM3 residues W311$^{3.50}$, A314$^{3.53}$, A315$^{3.54}$; TM5 residue L397$^{5.69}$; TM6 residues K413$^{6.25}$ and L414$^{6.26}$ (Supplementary Fig. 11). However, several Nb9-binding residues are unique to the FZD3/6 subgroup: TM6 residues N410$^{6.22}$,

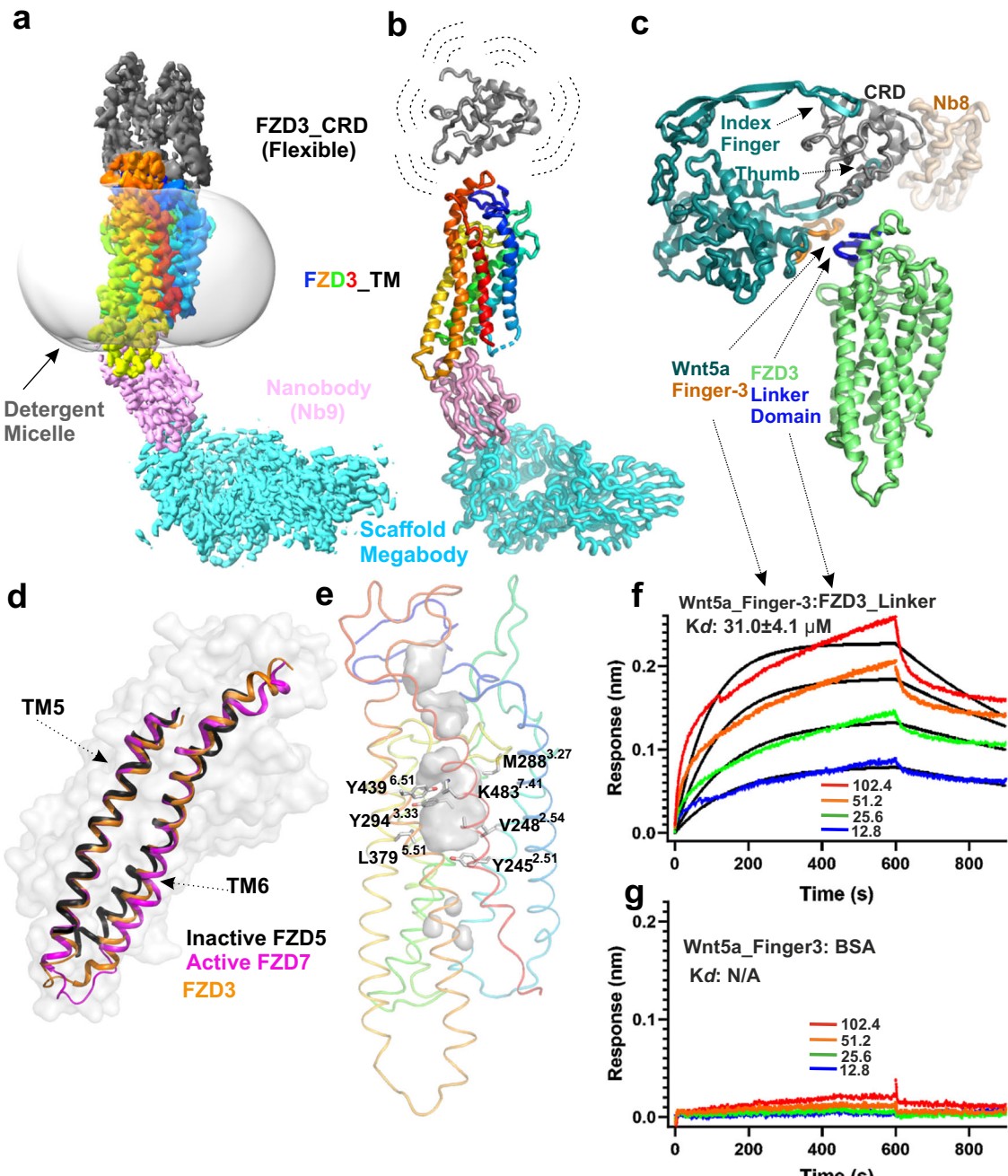

**Fig. 3 | The Cryo-EM structure of the FZD3 in complex with Nb9 megabody and BLI data for FZD3 linker peptide interacts with Wnt5a finger-3 peptide. a** The Cryo-EM map of FZD3 in complex with Nb9 Megabody. The extracellular region of FZD3 is coloured in dark grey, TM helices in rainbow, Nb9 in pink, megabody scaffold in cyan, and detergent disc in light grey (transparent surface). **b** Cartoon representation of the cryo-EM structure. The CRD is placed in the weak density showing a possible orientation with surrounding dashed lines indicating flexibility. **c** Superposition of a modelled Wnt5a-FZD3 CRD complex (Wnt3_FZD8, PDB 6AHY, was used as the template for modelling) and the FZD3 CRD–Nb8 complex structure (Nb8 coloured in wheat). The Wnt5a finger-3 is coloured orange and FZD3 linker domain is coloured blue. **d** FZD3 (orange) superimposed with the inactive FZD5 (PDB 6WW2, dark grey) and the active FZD7 (PDB 6EVW, magenta). Only TM5/6 are displayed for clarity. The FZD3 surface is shown for relative TM5/6 location. **e** Putative small molecule-binding pocket in the FZD3 TMD (grey surface). FZD3 residues that correspond to FZD7 residues proposed to interact with the small molecule F7H are indicated. Numbering of residues according to the Ballesteros-Weinstein scheme is shown in superscript. **f** BLI sensorgrams for biotinylated Wnt5a finger-3 peptide interacting with the FZD3 linker peptide. **g** BLI sensorgrams for biotinylated Wnt5a finger-3 peptide interacting with BSA (bovine serum albumin). Different concentrations are shown by coloured lines with indicated concentrations (μM). Fitted curves are shown by black lines.

Q411$^{6.23}$, and F417$^{6.29}$ (Supplementary Fig. 11). Our binding studies showed some cross-reactivity by Nb9 that is consistent with FZD3 and FZD6 sharing multiple Nb9-binding residues. Kd for FZD6-Nb9 interactions is $410 \pm 41$ nM (Fig. 1f), more than an order of magnitude weaker than that of FZD3 ($32 \pm 2.8$ nM, Fig. 1e). This difference may be attributed to unique FZD3-Nb9 interacting residues: ICL2 S321 (C in FZD6, H or N in other FZDs), which forms a hydrogen bond with Nb9 D31; and ICL3 E403 (V, I, or N in other FZDs), which interacts with Nb9 I27. The affinity of Nb9 for FZDs outside the FZD3/6 subgroup was not tested, but the additional differences at residues that mediate FZD3 Nb9 interactions suggest that other FZDs would exhibit lower affinities for Nb9 than that seen for FZD6.

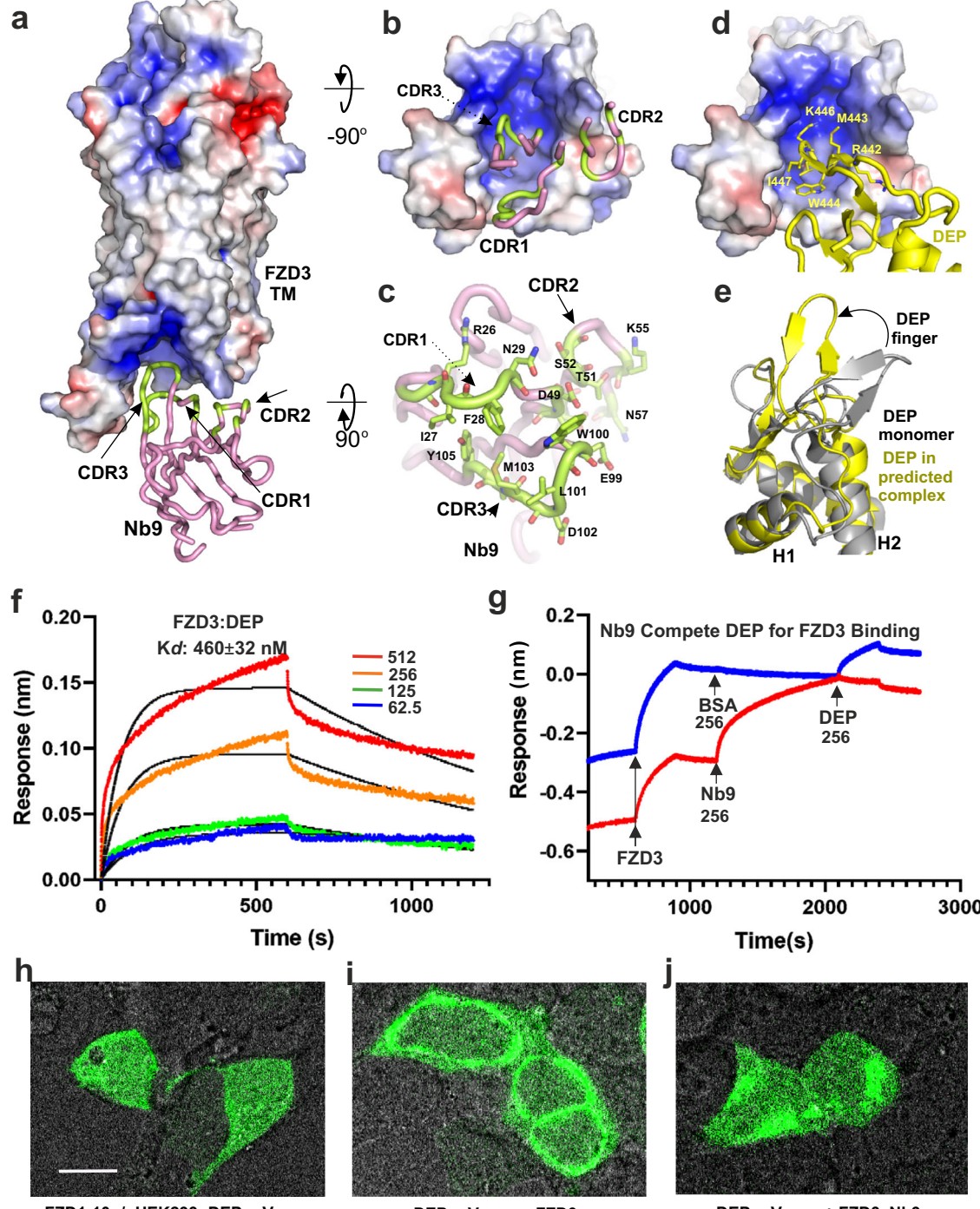

**Fig. 4 | Nb9 competes with DVL DEP domain for binding to FZD3. a** The structure of FZD3 transmembrane domain (as electrostatic surface) in complex with Nb9 (pink tube with the interacting CDR residues in lemon green). **b** Bottom view of FZD3, showing the Nb9 binding pocket. **c** Top view of the Nb9, with interacting residues shown as sticks. **d** AlphaFold2 Multimer prediction of the FZD3-DVL (DEP domain, yellow) complex. The labelled residues are known to be important for FZD binding. **e** The DEP domain from the predicted complex superimposed with the apo structure (PDB 1FSH). **f** BLI data showing the affinity of DEP domain (from DVL2) binding to FZD3. **g** Nb9 (256 nM, red line) competes with DVL2 DEP (256 nM) for FZD3 binding. BSA (blue line, 256 nM) is used as control. **h** Confocal image of DEP-monoVenus from transfected FZD1–10-/- HEK293T cells. **i** Transfection of DEP-monoVenus with FZD3. **j** Transfection of DEP-monoVenus with FZD3_Nb9. The scale bar is 10 μm. The corresponding percentages of cells showing membrane localization for **h**, **i** and **j** are shown in Supplementary Fig. 12b.

## Nb9 competes with DVL DEP Domain binding to FZD3

The DEP domain of DVL is responsible for FZD binding[16] through its hairpin loop (so called 'DEP finger'). Monomeric DEP binds to FZD while the domain swapped dimer hides the binding finger and triggers canonical signalosome assembly[49]. To visualise how FZD3 might interact with DVL, we used AlphaFold[50] to model the FZD3-DVL2 complex. The modelled complex shows the DEP finger binding into the concave surface of the FZD3 intracellular region using residues that have previously been shown to interact with FZD[49] (Fig. 4d). The predicted binding surface on FZD3 overlaps with the Nb9 epitope (Supplementary Fig. 12a). We note that the DEP finger is shifted upwards in forming this putative complex with FZD3 (Fig. 4e), rather than folding

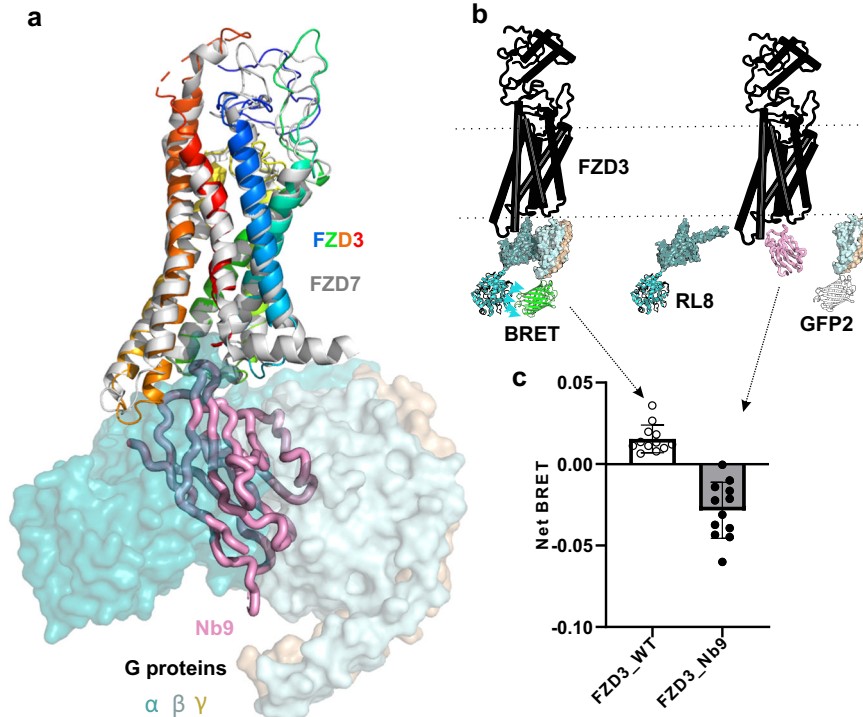

**Fig. 5 | Nb9 interferes with basal heterotrimeric G proteins coupling to FZD3.**
**a** FZD3 (rainbow cartoon) in complex with Nb9 (pink tube) superimposed with
FZD7 (grey cartoon) in complex with heterotrimeric G proteins (GαS in dark teal,
Gβ in light teal, and Gγ in wheat surface). **b** Schematic representation of biolumi-
nescence resonance energy transfer 2 (BRET2). FZD3 is in black, GαS linked with
Renilla luciferase 8 (RL8) in cyan, Gγ linked with green fluorescence protein 2

(GFP2) in grey or green to indicate BRET. **c** The Net BRET values of wild-type FZD3
(FZD3_WT, empty circles indicate each data value) and FZD3 linked to Nb9 at its
C-terminus (FZD3_Nb9, filled circles, bar with shading). The difference between the
two net BRET values is statistically significant ($n = 12$, three biological experiments
with quadruplicates, $P < 0.0001$, two-side t-test, the error bars are presented as
mean values +/-SD).

back on itself as in the structure of monomeric DEP domain (PDB
1FSH)[16].

Given the Nb9 epitope overlaps with the predicted DEP binding
site in FZD3, we performed a BLI competition assay. Firstly, we deter-
mined the affinity of DVL2 DEP domain for FZD3. The measured Kd was
$460 \pm 32$ nM (Fig. 4f), which is about 14-fold lower than that of Nb9 (Kd
$32 \pm 2.8$ nM). We then loaded two BLI tips with a similar amount of
biotinylated FZD3, dipped them into Nb9 or a BSA control, then tested
binding to DEP domain. As shown in the Fig. 4g, the FZD3-BSA control
shows normal binding to DEP domain. In contrast, the FZD3-Nb9
sensor tip showed negligible binding to DEP domain. This experiment
shows that Nb9 can block interactions between DEP domain and FZD3.

We further tested whether Nb9 can block interactions between
the DEP domain and FZD3 in living cells using a DEP domain recruit-
ment assay[49]. In HEK293T cells lacking all ten human FZD family
members (FZD1–10$^{-/-}$)[51], the DEP-monoVenus protein displays an even
distribution within cells (Fig. 4h), and does not form punctate because
only the DEP domain is present instead of full-length DVL2. When FZD3
is co-transfected, fluorescence shifts to the cell membranes, indicating
recruitment of the DEP-monoVenus (Fig. 4i). However, when the
cytoplasmic C-terminus of FZD3 is fused with Nb9, the fluorescence
signal at the cell membrane is greatly reduced (Fig.4j) with statistical
$P$ value $< 0.0001$ (Supplementary Fig. 12b). These experiments suggest
that Nb9 competes with the DVL DEP domain for FZD3 binding.

**Nb9 interferes with FZD3–heterotrimeric G protein interactions**
The FZDs and SMO together comprise the F class GPCRs, and can
signal via G proteins[52]. FZD3 was previously reported to be able to
signal through Gαs[53]. The structure of FZD7 in complex with Gαs, Gβ
and Gγ has been determined[13]. While this manuscript was under revi-
sion, structures of FZD1, 3 and 6 in complex with G proteins were also

reported[54]. Superimposition of our FZD3-Nb9 structure with the FZD7-
G protein structure (PDB 7EVW), shows that Nb9 overlaps with Gα and
Gβ (Fig. 5a). Therefore, to test if Nb9 can disturb the basal GαS, Gβ and
Gγ coupling to FZD3, we used bioluminescence resonance energy
transfer (BRET) Gαβγ biosensors (Gα linked with RLuc8 and Gγ linked
with GFP2) from the TRUPATH BRET2[55] (schematically shown
in Fig. 5b).

When FZD1–10$^{-/-}$ HEK293T cells were transfected with the TRU-
PATH triple GαsS plasmid and FZD3, a low basal net BRET value was
observed (net BRET is calculated from FZD BRET ratio minus a control
(pcDNA3) ratio in the same context, Fig. 5c). This small constant
positive value indicates that FZD3 may possess basal GαsβγS coupling.
However, when cells were instead transfected with Nb9 fused to the
C-terminus of FZD3, a negative BRET value was observed (Fig. 5c). This
difference is statistically significant ($P < 0.001$, t-test, $n = 12$) and sug-
gests Nb9 could disturb the basal GαS from forming a heterotrimeric G
protein complex. Notably, this lower BRET value contrasts with tradi-
tional GPCR activation BRET assays, where a lower signal can result
from agonist-induced disassociation from the coupled G protein sub-
units with GDP/GTP exchange events. Nb9 prevents the initial asso-
ciation step and thus may disrupt agonist-induced GDP/GTP exchange
required for GPCR activation.

## Discussion
FZDs are the primary receptors for Wnt morphogens and are essential
for Wnt signalling, which controls cell proliferation, differentiation,
migration, and polarity. An improved understanding of the molecular
mechanisms of FZD receptor activation or inhibition, and the devel-
opment of inhibitory or activating FZD modulators based on this
understanding, could potentially lead to new therapeutics for cancers
or degenerative diseases where Wnt signalling is mis-regulated.

Many FZD binders, especially antibodies, have been developed and show promising cancer therapeutic effects in animal models[30,56]. On the other hand, FZD binders, when linked to LRP5/6 binders, can serve as surrogate ligands that are able to mimic Wnt to boost stem cell regeneration[34,57]. Interestingly, a surrogate specifically targeting the non-canonical Wnt receptor FZD6 has been shown to substitute for Wnt signalling though the canonical receptor FZD5 in alveolar regeneration demonstrating the potential of surrogates to provide "pharmacological gain of function" effects[36]. Here, we have characterised two novel FZD3-specific nanobodies (the extracellular CRD binder Nb8 and the intracellular region binder Nb9) which may have potential for therapeutic applications after humanization and affinity maturation. We also report the design and functionality of a dimeric Nb8-based Wnt surrogate.

Several studies have pointed to the importance of receptor clustering for Wnt signalling[58–61]. Crystallographic analyses of several FZD CRDs prompted the suggestion that fatty acyl recognition mediates dimerization[10] and this dimerization interface is also present in the crystal structure of the FZD8 CRD-Wnt3 complex[37]. Superposition of our FZD3 CRD Nb8 complex onto the FZD8 CRD dimer indicates that Nb8 binding would hinder CRD-CRD interactions (Fig. 2d), potentially inhibiting Wnt signal transduction by interfering with FZD3 dimerization.

Our Nb8 based Wnt surrogate was engineered to have bivalent binding, specifically to FZD3 through the Nb8 module and to LRP5/6 through the DKK1C module. The Nb8-DKK1C fusion comprises the minimum set of interaction surfaces required for FZD3 - LRP6 heterodimerisation. For favourably positioned binding epitopes this bivalent architecture has provided potent Wnt surrogates[35]. Additionally, we included a C-terminal Fc fusion as a putative promoter of multivalent binding and hence FZD3 dimerization. Similarly to other bispecific tetravalent Wnt surrogates[39,57,62], the Nb8-based surrogate initiates β-catenin signalling. Alongside the recently reported FZD6 specific surrogate[36], the activity of our FZD3 specific surrogate demonstrates that members of the "non-canonical" FZD3/6 subset also harbour the ability to trigger the canonical FZD signalling pathway.

DVL is a pivotal intracellular mediator of Wnt signalling (canonical and non-canonical) that uses its DEP domain to bind FZD[63]. Monomeric DEP can bind to FZD in the absence of extracellular Wnt binding. Our results show that Nb9 can compete with the DEP domain of DVL for FZD3 binding (Fig. 4g). FZD3 has also been reported to be able to signal through Gαs proteins[53]. Our BRET2 assay demonstrates that FZD3 has basal Gαs, β, γ heterotrimer G protein coupling, and that Nb9 can interfere with this coupling. Indeed, the central positioning of the Nb9 binding epitope in the concave surface of the FZD3 intracellular region suggests that it may serve as a valuable tool for studying downstream signalling mechanisms.

We engineered Nb9 to generate a megabody that facilitated the structure determination of full length FZD3. The FZD3-megabody complex yielded single particle cryoEM density for the extracellular region that allows us to orient the CRD relative to the well-defined helices of the TMD. Linker sequences vary substantially among FZDs (Supplementary Fig. 8a) and may reflect differing functional requirements for CRD orientation and flexion. The CRD orientation we observe for FZD3 may be tuned to facilitate Wnt binding and/or co-receptor recruitment for non-canonical signalling. In contrast, the flexibly linked CRD in the FZD5-xWnt8 complex samples orientations that are rotated by some 90° relative to the CRD in FZD3, possibly enabling LRP5/6 binding[12].

In humans there are 19 Wnt ligands and 10 Frizzled receptors. How different Wnts discriminate between FZDs to initiate different signalling regimes remains poorly understood. The interaction of Wnts (using their so-called thumb and index finger) with FZD CRDs is the primary source of pathway selectivity[6]. However, given the high degree of conservation in Wnt residues involved in FZD CRD binding (Supplementary Fig. 8b, c), as well as the binding sites on the FZD CRDs themselves, these interactions cannot account for all Wnt-FZD specificity. Recently, the FZD linker domain between the CRD and TMD domains has been identified as participating in Wnt ligand recognition[7]. However, this raises the question of which part of the Wnt ligand is responsible for binding to the linker domain. Our full length FZD3 structure with CRD orientation, modelling of the FZD3-Wnt5a complex and the FZD3 linker domain Wnt5a finger-3 binding data, suggest a Wnt finger-3 could be suitably positioned to interact with the FZD linker domain. Whether such interactions contribute to the specificity of all the Wnt-FZD recognition remains to be fully explored. However, our data suggests that investigation of the Wnt finger-3 and FZD linker domain interactions is warranted. More generally, the nanobody modulators and structural data we present here provide the information and tools with which to specifically target FZD3 biology and therapeutic potential.

## Methods

### Discovery of the nanobodies

One llama (*Lama glama*) was immunized with purified recombinant FZD3 in 10 mM HEPES pH 7.5, 150 mM NaCl, 1% glycerol, 0.03% DDM, 0.006% CHS. All selections were done in the same buffer. Panning was performed on 1 µg of FZD3 or 1 µg of FZD3 TMD coated on maxisorb wells or on neutravidin-captured biotinylated FZD3 (400 nM). FZD3 specific phages were recovered by incubating the coated wells with Trypsin (250 µg/ml) for 30 min. Trypsin was neutralized by adding AEBSF and phages were added to freshly grown TG1 cells. After 2 rounds of selection, individual colonies from all selection conditions were screened for the expression of FZD3 specific nanobodies in an ELISA. For the detection of the presence of specific Nanobodies, the "EPEA"-tag at the end of the Nanobody was used (LifeTechnologies #7103252100) in combination with Streptavidin Alkaline Phosphatase (Promega #V5591). Positives were detected by using 4-nitrophenyl phosphate disodium salt hexahydrate (2 mg/ml). The absorption at 405 nm was measured for 60 min after adding the enzyme substrate p-nitrophenyl phosphate. 171 clones were sequence analysed to reveal enriched families. Family 1 (Nb1, 3-6) was identified 74 times, family 2 (Nb8) 39 times and family 3 (Nb9) 5 times. Family 3 was identified as a binder to the TMD.

Nb8 for binding experiments was produced in mammalian cells. For purifying Twin-Strep tagged FZD3 CRD-Nb8 complexes, the conditioned media were passed through a Strep-Tactin XT Superflow 5 ml high capacity cartridge (IBA). Following a wash step with the washing buffer (25 mM Tris pH 8.0, 150 mM NaCl, 1 mM EDTA), the resin was then incubated overnight at 4 °C with 1 column volume of wash buffer containing Rhinovirus 3C protease (-1 mg/ml) at a dilution of 1:2500 (v/v) and endoglycosidase F1 (-1 mg/ml) at a dilution of 1:1000 (v/v) to liberate the FZD3 CRD-Nb complexes for elution and to deglycosylate the sample. Size exclusion chromatography (SEC) was then performed using a Superdex 75 10/300 GL column (GE Healthcare) equilibrated with 10 mM HEPES pH 7.5, 50 mM NaCl. The Nb9 megabody was produced in bacteria (WK6 strain) with the pMESP23E2 vector using a previously published procedure[42]. Briefly, the nanobody was PCR amplified with the primers: TU89 and EP230 (see Supplementary Data 1 for oligo sequence). The PCR product was digested with SapI and ligated with SapI/SalI double cut pMESP23E2 vector. Transformed WK6 bacteria were used for megabody production after a sequencing check. The resulting expressed protein contains (from the N-terminus) the first β-strand of the nanobody (AA 1-13, an universal sequence for all nanobodies embedded in the vector), YgjK (AA24-783, with a flexible linker in the middle), and ended with the nanobody sequence starting SLRLSC (i.e. omitting the first β-sheet), as previously described[64].

### Protein expression

Wild type full length human FZD3 (UniProt ID Q9NPG1, amino acid residues 1-666) was used for functional assays. The C-terminally

truncated (residues 26-517) FZD3 construct tagged at the C-terminus with a Rhinovirus 3C cleavage site, monoVenus fluorescent protein, and a Twin-Strep tag, was used for multidomain protein production. For FZD3 CRD production, the construct contained residues 26-138 and was tagged in the same manner. All constructs were cloned into a lentiviral vector, and lentiviral transduction of HEK293S GnTI-cells was performed as described previously[65], or directly used for transient transfection without a lentiviral step. For producing biotinylated proteins for BLI experiments, a HEK293T cell line was created constitutively expressing ER-localised biotin ligase[66]. The transfected plasmids (for either FZD3/6 C-terminal truncated transmembrane or just CRDs) with C-terminal Avi3 and 6×His-tag[67] were used for in vivo biotinylation, in which the culture media was supplemented with 100 μM biotin. For crystallization experiments, tagged FZD3 CRD and untagged Nb8 were co-expressed using lentiviral transduction in HEK293S GnTI- cells adapted to suspension culture. For cryo EM membrane protein production, HEK293F cells in suspension culture were used. For purifying FZD3, the lentivirally transduced HEK293F cells were grown in suspension. Harvested cell pellets were solubilised in a buffer comprising 100 mM Tris-HCl, pH 8.0, 150 mM NaCl, 1% LMNG (Lauryl Maltose Neopentyl Glycol, Anatrace), 0.05% cholesteryl hemisuccinate (CHS, Sigma), and protease inhibitor mixture (Roche), for 2 hours at 4 °C with rotation. After a centrifugation step (35,000 × g for 30 min), the supernatant was passed through a Strep-Tactin XT cartridge. After washing, the cartridge was incubated with a buffer exchange solution (10 mM HEPES pH 7.4, 150 mM NaCl, 0.5% GDN (glyco-diosgenin), 0.02% CHS and protease inhibitor mix) for 2 hours at 4 °C and finally eluted in the same buffer supplemented with a 1/10 volume of 10× elution buffer (IBA). The proteins were further purified by SEC using a Superose 6 Increase 10/300 column (GE Healthcare), equilibrated with 10 mM HEPES pH 7.4, 150 mM NaCl, 0.05% GDN, 0.002% CHS, and protease inhibitor mix.

## Protein crystallisation

FZD3 CRD-Nb8 complexes were concentrated to 6–8 mg mL$^{-1}$ for crystallization. Sitting drop vapour diffusion crystallisation trials were performed whereby 100 nl of protein solution was mixed with 100 nl of reservoir solution using a robot (Cartesian Technologies) in 96-well crystallisation plates (Greiner). Crystals appeared in a condition containing 100 mM citric acid, pH 3.6 and 690 mM ammonium sulphate, and were cryoprotected using reservoir solution supplemented with 30% glycerol, prior to flash freezing in liquid nitrogen.

## Crystal data collection and structure determination

X-ray diffraction data were collected at the Diamond Light Source (Oxfordshire, UK) using the I03 beamline with wavelength 0.97625 Å at temperature of 100 K (−173 °C) and processed using Xia2. The structure was determined by molecular replacement using the mouse FZD8 CRD (PDB 1IJY) and a llama-derived Nb against the J-base binding protein 1 (PDB 6FPV) as search models in Phaser. PHENIX and Coot were used for refinement and model building. There are two essentially identical complexes in an asymmetric unit. The data collection and refinement statistics are presented in Supplementary Table 1, with favoured Ramachandran values of 98.13% and no outliers.

## Cryo-EM sample preparation and data collection

Fresh FZD3 protein at 2 mg mL$^{-1}$ was mixed with Nb9 megabody at an equal molecular ratio and incubated on ice for 30 min. 3 μL of protein mixture were applied onto copper grids (Quantifoil, R1.2/1.3, 300 mesh), and excess liquid was removed by blotting for 3 s (blot force -1) before plunge freezing in liquid ethane using an FEI Vitrobot Mark IV at 100% humidity and 4 °C. The grid was initially examined with a Glacios 200 kV Microscope before data were collected with a Titan Krios 300 kV electron microscope equipped with an energy filter and K3 camera (Gatan) at Diamond eBIC. The collection details are shown in

the Supplementary Table 2. The data were processed with CryoSPARC v4 and finally refined with 3DFlex[44]. The model was built with Coot, starting with an initially fitted FZD7 model (PDB 7EVW) and refined with PHENIX. The refinement statistics are shown in the Supplementary Table 2.

## Biolayer interferometry

Quantitative analyses of the interactions between nanobodies, DEP (produced by bacterial expression) or Wnt5a (purchased from Biotechne R&D systems, cat# 645-WN-010/CF), and biotinylated FZD3/6 CRD, or multidomain FZD3 (produced by in vivo biotinylation in HEK293T cells with transfection of Avi-tagged plasmid and biotin ligase birA expression plasmid) in detergent, were performed using an Octet Red 96e system (ForteBio). Proteins were diluted in a buffer containing 10 mM HEPES, pH 7.5, 150 mM NaCl, 0.02% Tween 20 and 0.005% GDN detergent). For testing theWnt5a finger3 FZD3 linker domain interaction, Wnt5a G163-C182 peptide was synthesized (by GenScript Biotech) with G163 biotinylation and C164-C182 cyclised by a disulfide bond. FZD3 liker C165-C186 was also synthesized with disulfide cyclisation. BLI curves were fitted using a 1:1 binding model and Kds were calculated using ForteBio Data Analysis software. For the initial qualitative testing of the 10 FZD CRDs binding to Nb8, a BLItz System (ForteBio) was also used. We note that the measured affinity between DEP and FZD3 in detergent is relatively low (Kd about 0.5 μM). As we were comparing FZD protein-protein interactions with Nb9 and the DVL DEP domain, our membrane protein preparation didn't include the lipid PI(4,5)P2, which had been shown to improve the binding between DEP and FZD4[68].

## TOPFlash luciferase assays

The FZD3 surrogate (Nb8-DKK1C) was constructed by fusion of Nb8 (residues 1-119) to DKK1 (177-266) with flexible linker sequence of SGSGSG and cloned in pHL-FcHis vector which contains human IgGγ1 Fc and 6×His at the C-terminus[67]. Luciferase assays were performed using FZD1–10$^{-/-}$ HEK293T cells[51] (gift from Prof. Benoit Vanhollebeke) in 96-well white back plates (Thermo Fisher Scientific). 4×10$^4$ cells were seeded per well in 100 μL of DMEM supplemented with 10% FBS, and kept in a humidified incubator at 37˚C, with 5% CO$_2$. After 24 hours, cells were transfected with various combinations of plasmids (30 ng per plasmid per well) including Super TOPFlash firefly luciferase[69] and tk-Renilla luciferase plasmid (Promega), with the total amount of DNA adjusted to 120 ng with empty vector (pcDNA3). Transfections were performed using TransIT (Merck) according to manufacturer's recommendations. The firefly and Renilla luciferase activities were measured 24 h later using the Dual-Glo luciferase reporter assay system (Promega) with an Ascent Luminoskan luminometer (Labsystems). As a positive control, STF 293T (ATCC, American Type Culture Collection, CRL-3249) cells were seeded in same concentration and induced with L-Wnt-3A cell (ATCC CRL-2647) conditioned media 24 hours before the assay. The firefly luciferase activity was normalized with the constitutive Renilla luciferase activity.

## BRET2 TRUPATH assay

The BRET2 Trupath assay was employed to test the role of Nb9 in interfering with constitutive G protein coupling to FZD3. FZD1–10$^{-/-}$ HEK293T cells were initially plated in a 6-well plate and transfected 24 hours after seeding with a control (pcDNA3), FZD3_WT (wild type, full length), or FZD3_Nb9 (Nb9 fused at the C-terminus of FZD3 with flexible linker) plasmid respectively with TRUPATH Triple GasS (Addgene#196055, encodes a G alpha subunit GNAS2 with RLuc8, a G gamma subunit, GNG9 with GFP2 and a G beta subunit, GNB3). The plasmid ratio of FZD to Trupath triple is 2.5:1, and TransIT was used according to the manufacturer's recommendations. 24 hours later, cells were transferred into a 96-well white backed microplate with a replicate in a 96-well transparent bottom tissue culture plate for

monitoring cells. 100 μl containing $5 \times 10^5$ cells for each well were seeded and incubated for an additional 24 h at 37 °C. For the constitutive BRET measurement, the white backed plate medium was replaced with medium supplemented with 5 μM coelenterazine 400a (Cambridge Bioscience Ltd). BRET luminance was detected with a CLARIOstar plate reader with 410–480 nm (RLuc8-coelenterazine 400a) and 515–530 nm (GFP2) emission filters. The GFP2 emission to RLuc8 emission ratio was further deduced by the ratio from the control plasmid (no FZDs) transfected cells. We note the overall BRET value measured in our assays was relatively small, which may be due to the CLARIOstar plate reader used, which has been reported to give a 100-fold lower BRET value than the Envision plate reader[70].

## Statistical analysis

GraphPad Prism 10 was used for data analysis of the TOP-flash luciferase assay, DEP recruitment assay and G-coupling block assay. A two-tailed unpaired Student's $t$-test was used for single comparison, $p < 0.05$ was considered statistically significant.

## Reporting summary

Further information on research design is available in the Nature Portfolio Reporting Summary linked to this article.

## Data availability

The FZD3 CRD -Nb8 complex crystal data generated in this study have been deposited in the Protein Data Bank (PDB) under accession code 8Q7O. The cryo-EM maps for the FZD3-Nb9 complex have been deposited in the Electron Microscopy Data Bank (EMDB) under accession code EMD-18680 and the PDB under accession code and 8QW4. The source data underlying all Figures and Supplementary Figs. are provided as Source Data file. Source data are provided with this paper.

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

## Acknowledgements

The project is supported by the Wellcome collaboration grant: 223133/Z/21/Z (E.Y.J.), the Wellcome Trust Core Award Grant: 203141/Z/16/Z (Centre for Human Genetics) and Cancer Research UK: C375/A17721 (E.Y.J.). and DRCRPG-May23/100002 (T.M.). We acknowledge the support and the use of the resources of Instruct-ERIC (PID6383 VID12131 to E.Y.J.), part of the European Strategy Forum on Research Infrastructures (ESFRI), and the Research Foundation - Flanders (FWO) for their support to the Nanobody discovery and the Megabody plasmids, and thank Eva Beke, Alison Lundqvist, Nele Buys and Katleen Willibal for the technical assistance during Nanobody discovery. We thank Professor Benoit Vanhollebeke for the FZD1–10$^{-/-}$ HEK293T cell line. We thank Dr. James Bancroft for his assistance with confocal microscope imaging. We thank Sir David I. Stuart for his valuable comments and editing. We thank Instruct-ERIC centres Diamond eBIC (Dr. Julika Radecke) and Strubi OPIC for EM data collection assistance. We thank Diamond beamline I03 (under BAG MX-19946) for crystal data collection and eBIC (under BAG bi-28713) for cryo-EM data collection. The cryo-EM data processing was performed using the Oxford Biomedical Research Computing (BMRC) facility, a joint development between the Wellcome Centre for Human Genetics and the Big Data Institute supported by Health Data Research UK and the NIHR Oxford Biomedical Research Centre.

## Author contributions

J.H. designed and optimised FZD3 expression constructs and purification strategies, produced FZD3 CRD and nanobody samples, performed crystallization, structural determination and performed functional assays. Y. Z. purified full length FZD3 protein as well as Nb9 megabody, and determined cryo-EM structure of the complex, performed functional assays. L. C. and T. N. provided cryo EM expertise. T.M. performed molecular replacement and AlphaFold modelling. R.R.R. performed nanobody selections and provided DVL constructs. T.C. started membrane protein set up. G.Y. H.M.E.D., J.Y., Y.Z. participated in cryo-EM data

collection, analysis, discussion and manuscript improvement. W. L. performed cell culture and transfection for protein production. E.P. and J.S. performed and supervised the Nb discovery. E.Y.J. designed the project. E. Y. J., Y. Z., J. H. and T.M. wrote the manuscript.

## Competing interests

The authors declare no competing interests.
