## [Transparent Peer Review file · Nature Communications]

Structural Insights into Frizzled3 through Nanobody Modulators

Corresponding Author: Professor Edith Yvonne Jones

Figures originally included in the author's rebuttal have been redacted from this file.

Version 0:

Reviewer comments:

Reviewer #1

(Remarks to the Author)

The manuscript presents a range of structural snapshots of Frizzled3: CRD complex with Nb, full length complex with several chaperone Nbs and Ab that allow some semblance of the ECD to be resolved partially, and finally Fzd3 complexed with an inhibitory Nb that binds to the intracellular regions and blocks signaling adaptors from binding. In the case of the CRD binding Nb the authors linked it to DKK and showed that it functions as a surrogate agonist, analogous to previous work. A Wnt control is missing from the assay so we have no idea what magnitude of signal is being generated and this is very important to know.

The work is solid, well done technically and contributes to the growing database of Fzd structures and complexes with various ligands that collectively constitute a toolbox of structures that can be accessed by the Wnt community to interrogate mechanism and design drugs, this seems to be the most direct contributions of the current work, as it lacks much new biological insight that changes or modifies how we think about Wnt/Fzd structural biology.

Reviewer #2

(Remarks to the Author)

In their manuscript entitled "Structural Insight into Frizzled3 through Nanobody Modulators", Hillier et al. identify and structurally characterize two FZD3-binding nanobodies, a CRD-binding Nb8 and an intracellularly binding Nb9. Authors use these nanobodies to gain insights into FZD receptor biology, specifically the mechanism of Wnt binding and activation.

While it is pretty clear and well justified that authors did indeed produce FZD3 (and FZD6) binding nanobodies that do indeed bind to a CRD region (Nb8) and at the intracellular site (Nb9), the utility of these nanobodies to understand the biology of FZD receptor is limited. Most conclusions about how Wnt binds and activates FZD receptors are based on their cryo-EM structure of FZD3-Nb9 (where they claim to see enough of the CRD domain density to place their x-ray structure). However, that density is of poor quality, and the experimental data does not support many of the authors' claims (see detailed list below). Without that, the paper essentially becomes the characterization of two FZD nanobodies. With that in mind, the data is not very exciting and quite predictable. Nb8 binds to CRD and might interfere with Wnt binding (plenty of biologics that bind CRD and interfere with Wnt binding have been described for other FZD receptors already- e.g Diamond, J. R. et al. *Breast Cancer Res Treat* 184, 53-62 (2020); Li, K. et al, *Cancer Sci* 114, 2109-2122 (2023), Pavlovic, Z. et al. *MAbs* 10, 1157-1167 (2018), etc.). Nb 9 binds intracellularly and, as expected, competes with Dvl and G protein binding. With that in mind, the other novel part might be the structure of the FZD3 receptor. And, while the CRD part (from the x-ray structure) is of excellent quality, the TM part (from cryo-EM) is not. Particularly worrisome is the very loose placement of ECL loops into the density, which is clearly noise from the micelle. Since this might be the first deposited FZD3 structure, it would be prudent to exercise caution and conservatism with modelling and omit the regions that are not observed (more comments below).

The most exciting finding was that the forceful coupling of FZD3 to LRP5/6 through Wnt surrogates promotes β -catenin signalling in the receptors that are not known for it.

While there are some parts of the manuscript that are quite interesting, many aspects are rushed to conclusion at the expense of the experimental evidence. In my opinion, this manuscript needs to be reworked to omit the parts in question. Unfortunately, without them the story becomes less exciting.

Major issues:

1. Placement of the CRD domain into the density. A fairly significant portion of the manuscript is dedicated to identifying the CRD region in the cryo-EM map of FZD3-Nb9 and the implications of such CRD placement for Wnt binding. However, I question the validity of this structure (see also the following comment). While there is no doubt that authors have a CRD domain somewhere in that overall region in their cryo-EM map, it is still incredibly flexible. The density is very fuzzy and does not correlate to the determined FZD3 CRD structure. While the overall region appears correct, there is no way to orient the CRD pdb in any particular way to fit the density with confidence. The fit would be equally good (and wrong at the same time) if it was rotated 180 ° or upside down. The current model does not have a good model-map correlation. Most likely, the observed “fuzz” of the CRD domain represents an amalgamation of many unresolved conformations (or possibly even noise-see below), making any predictions about its orientation relative to the TM unfounded. This information then also invalidates any predictions about Wnt binding at the full-length FZD receptor (Fig. 3c) and any discussion about the role of the linker in Wnt binding and recognition. This entire section “The FZD3 extracellular domain and Wnt specificity” is not supported by experimental data.
2. The quality of the cryo-EM data is quite problematic. The FSC curves (Fig. 6) show an incomplete drop to 0 for both 3D Flex and the Local refinement. The latter is particularly worrisome as the FSC has an additional shoulder that brings the resolutions to 2.94Å, far above what one expects from the curves with that drop to 0 as expected. This shoulder is most likely due to the correlation of the noise at higher frequencies or a subset of bad particles present in the dataset. However, the noise explanation is more likely considering the quality of the maps, particularly in the ECD regions. The map does not correlate to the model, and most of the loops are missing from the EM density despite being present in the model. The incompleteness and the lack of density for the loops is quite severe, and even the backbone is untraceable in most of ECL1, ECL2 and ECL3. Notably, the conformation of ECL1 and ECL3 is completely different even between the two structures authors used themselves- the one from consensus refinement (and presumably deposited in the PDB) and the one used to place CRD from the 3DFlex refinement.
As an extension of this issue, I wonder whether the observed CRD density is also due to noise correlation, considering that the FSC does not drop to 0 for that structure as well and the fact that the backbone of the CRD is untraceable and does not resemble even loose correlation to the map.
3. The BLI experiments showing the competition of Nb8 and Wnt5a are very confusing (Fig. 1d). Why did authors need a dissociation step following Nb8 binding to FZD3 CRD if they intend to look at the competition between Wnt5a and Nb8? Why is the dissociation of Nb 8 not complete here, even though it appears to come off completely in Fig.1a? The difference between Wnt5a binding to Nb8 bound and unbound FZD3 CRD does not look like much. How did authors determine that “the Nb8-dipped tip showed reduced binding to Wnt5a, indicating that Nb8 and Wnt5a compete for FZD3 binding”?

Minor issues:

1. Representation of the 3DFlex map colored by local resolution (figure 6 middle right). Currently the range for the coloring is 2.5-3.5 Å, which misleadingly represents the entire CRD domain as having resolution at 3.5Å. This range needs to be extended significantly lower to showcase the true resolution of that region (perhaps to the range of calculated during the local resolution job).
2. What is the source of Wnt5a? and how was it prepared?
3. Add labels to the concentrations on Fig.1
4. Some ligands in the x-ray model have hydrogens (e.g. glycerol). Please be consistent.
5. Please state how was FZD3 prepared for immunization (buffer? Detergent? Nanodiscs? Etc.).
6. How was cryo-EM map sharpening done?
7. Describe how masks were generated for 3DFlex and local refinement.
8. There is a weird density (or abundance of it) near the intracellular parts of TM5 and 6. Please comment on this.

Reviewer #3

(Remarks to the Author)

The authors present an in-depth structural analysis of two nanobodies targeting the FZD3 receptor using multiple structural techniques, crystallography and cryo-EM. The binding interfaces of both an intracellular and extracellular nanobody were resolved with residue level precision, which will facilitate further tool development. Nb8 binds to the CRD region of FZD3, while Nb9 binds to the putative DVL and G protein binding site. The engineered Nb8 (Nb8-DKK1C) is able to induce activation of the canonical signaling pathway, while FDZ3 is known to activate exclusively to non-canonical signaling pathways.

Although Nb8, the engineered Nb8 and Nb9 are interesting tools, the structures, biophysical and cellular assays presented in this manuscript do not, as suggested, provide insight into the mechanisms of FZD3 signaling. Moreover, data on the mode of action of the nanobodies as modulators on basal and Wnt5a-induced signaling in the non-canonical and canonical signaling assays is limited or lacking.

1. The current data show that Nb8 binds to the site Wnt5a is binding to as it impairs binding of Wnt5a to FZD3 as shown in the BLI studies. FZD3 is reported to signal exclusively through non-canonical Wnt pathways, without eliciting β -catenin-dependent signaling. Does Nb8 inhibit the functional responses induced by Wnt5a in non-canonical signaling assays?
2. The engineered Nb8-DKK1C nanobody appears to induce a canonical response, activating TOP flash reporter activity. Although this receptor has reported not to signal to this pathway include Wnt5a as a (negative) control. What effect does this engineered Nb8 elicit in the non-canonical β -catenin assay(s)?

3. It would be informative to provide a more clear illustration (Fig.2e) of the Nb8-DKK1C nanobody fusion protein.
4. The structural superposition of FZD3 CRD Nb8 complex onto the FZD8 CRD dimer indicates that Nb8 binding would hinder formation of the CRD dimerization interface (Fig. 2d). Is there any evidence FZD3 CDR exists as dimer or dimerization is important?
5. When FZD3 is co-transfected with DEP-mVenus, fluorescence is mainly restricted to the plasma membrane, indicating basal association of the DEP-mVenus (Fig. 4i). Can this be modulated by Wnt ligands, Nb8 or Nb8-DKK1C?
6. FZD3 was previously reported to be able to signal through G α s. In Fig. 5C, a decrease in the BRET ratio between G and G is expected for G protein activation. Therefore, the results shown seem to suggest that FZD3 interferes with basal G protein activation or coupling and instead promotes reassociation when unobstructed. I would not call this coupling in a traditional GPCR sense. Signaling by Wnt5a should be taken along as a control.
7. Why is the structure of FZD3 superimposed in Fig 3d onto the active and inactive structure of SMO and not the constitutively active FZD7–mGs, and inactive structures of FZD4 and FZD5?
8. On pg. 10, RMSD is presented twice without indication of what atoms are being compared. Additionally, the authors indicate that there is only a 0.2A difference in RMSD when comparing their complex to active or inactive structures. From this analysis, it seems difficult to conclude anything regarding the similarity of the presented complex to previous structures as is claimed in the following line. Comment on why the RMSD analysis gives such similar results for both conformational comparisons or alter the analysis appropriately.
9. Nanobodies were selected using phage display. Interestingly, this resulted in 5 out of 7 lead molecules with almost identical sequences. It would be interesting to the field to provide more information on how these 7 were selected, amount of receptor used for panning and on which basis the 7 molecules were selected. How many rounds of panning, clones picked and how often did the individual sequences appear amongst these? This might reveal information about a potential bias towards specific epitopes on FZD3.
10. For the cross reactivity check on the different FZD CRDs, the authors nicely checked all 10 CRDs. Based on Suppl Fig 4 and the binding data, can the authors deduct a potential epitope from the cross-reactivity with FZD6?
11. Binding of Nb1 to full length FZD3 was tested but affinities were not calculated. Given that this is the largest cluster of sequences in Suppl Fig 1, it might be assumed that Nbs1-7 actually have the highest binding affinity towards FZD3. Can the authors elaborate on how these relatively low signals in BLI (less epitopes available) and the relatively high abundance in panning outputs?
12. In line 136, the authors indicate the contact surface between Nb8 and FZD3 CRD. How do these correlate to those from other Nb-antigen interfaces? And can the authors elaborate on the nature of the interactions in this surface and how this correlates to the relatively high affinity? Like the Methionine?
13. Line 145. Can the authors specify the interactions partners of M116 and K112?
14. Line 193. Limited information is available on how the megabody was designed. In between which residues was the Ygjk inserted and how was this construct characterized. Did this insertion affect the binding affinity Nb9?

Additional points

15. The binding affinities (lines 109 and 111, 121-124, lines 282 and 283) should be indicated with SD or SEM and N.
16. The superimposition of the CRDs mentioned on pg 6 should be shown somewhere or the figure needs to be called.
17. Line 162, CDR dimerization should be CRD dimerization.
18. Pg. 19, GDN is not defined in text.
19. In Fig. 1, the binding curves in e and f are poorly fit and perhaps the model used needs to be reconsidered.
20. Fig. 4f, the binding curves are clearly biphasic and poorly fit with the 1:1 model used.
21. Fig. 5, the title states "... a Nb8-based Wnt surrogate activates canonical signaling", however this data was not presented in the figure or elsewhere.
22. Supp. Fig. 3: It is unclear what the numbers in the legend are indicating and panel b has no legend. Additionally, the Nb9 traces in panel b are all the same shade of gray and indistinguishable.
23. Supp. Fig. 5: The magenta used for Nb8 in panel b should be changed so as to make the red oxygen atoms easier to see.
24. The authors inconsistently use British and American spellings, sometimes within the same sentence. This needs to be addressed.

Author Rebuttal letter:

Reviewer #1 (Remarks to the Author):

The manuscript presents a range of structural snapshots of Frizzled3: CRD complex with Nb, full length complex with several chaperone Nbs and Ab that allow some semblance of the ECD to be resolved partially, and finally Fzd3 complexed with an inhibitory Nb that binds to the intracellular regions and blocks signalling adaptors from binding. In the case of the CRD binding Nb the authors linked it to DKK and showed that it functions as a surrogate agonist, analogous to previous work. A Wnt control is missing from the assay, so we have no idea what magnitude of signal is being generated and this is very important to know.

The work is solid, well done technically and contributes to the growing database of Fzd structures and complexes with various ligands that collectively constitute a toolbox of structures that can be accessed by the Wnt community to interrogate mechanism and design drugs, this seems to be the most direct contributions of the current work, as it lacks much new biological insight that changes or modifies how we think about Wnt/Fzd structural biology.

Response: We appreciate the reviewer's appraisal of our manuscript and the constructive comments. We have now added Wnt controls for the surrogate agonist assay, updated Fig.2f, and discussed its strength with comparison of the other published antibody-based surrogates. In page8, it reads: "The signalling level by our Nb8_DKK1C surrogate is comparable to that of antibody-based FZD6-targeting Wnt surrogates (ref#38) where it is sufficient to be functional. However, the overall measured TOP FLASH level by our Nb8_DKK1C surrogate is far lower compared to the level induced by Wnt3a-containing conditioned media in STF (SuperTopFlash) (ref#39) cells containing multiple FZDs (Fig. 2f)."

And added the corresponding method details in the end of the methods section of "TOPFlash luciferase assays", page 24, it reads: "For positive control, STF 293T (ATCC, American Type Culture Collection, CRL-3249) cells were seeded in same concentration and induced with L-Wnt-3A cell (ATCC CRL-2647) conditional media 24 hours before the assay."

For the comment about new biological insights, we have now included experimental data showing the FZD3 "linker" domain interacts with Wnt5a finger-3, which further validates our structural model of FZD3 CRD orientation. For the first time, it shows Wnt "finger-3" (apart from the two other well-known index finger and thumb) is also involved in FZD interaction through the "linker" domain. We believe this discovery casts new light on Wnt/FZD interaction.

Reviewer #2 (Remarks to the Author):

In their manuscript entitled "Structural Insight into Frizzled3 through Nanobody Modulators", Hillier et al. identify and structurally characterize two FZD3-binding nanobodies, a CRD-binding Nb8 and an intracellularly binding Nb9. Authors use these nanobodies to gain insights into FZD receptor biology, specifically the mechanism of Wnt binding and activation.

While it is pretty clear and well justified that authors did indeed produce FZD3 (and FZD6) binding nanobodies that do indeed bind to a CRD region (Nb8) and at the intracellular site (Nb9), the utility of these nanobodies to understand the biology of FZD receptor is limited. Most conclusions about how Wnt binds and activates FZD receptors are based on their cryo-EM structure of FZD3-Nb9 (where they claim to see enough of the CRD domain density to place their x-ray structure). However, that density is of poor quality, and the experimental data does not support many of the authors' claims (see detailed list below). Without that, the paper essentially becomes the characterization of two FZD nanobodies. With that in mind, the data is not very exciting and quite predictable. Nb8 binds to CRD and might interfere with Wnt binding (plenty of biologics that bind CRD and interfere with Wnt binding have been described for other FZD receptors already- e.g Diamond, J. R. et al. *Breast Cancer Res Treat* 184, 53-62 (2020); Li, K. et al, *Cancer Sci* 114, 2109-2122 (2023), Pavlovic, Z. et al. *MAbs* 10, 1157-1167 (2018), etc.). Nb 9 binds intracellularly and, as expected, competes with Dvl and G protein binding. With that in mind, the other novel part might be the structure of the FZD3 receptor.

And, while the CRD part (from the x-ray structure) is of excellent quality, the TM part (from cryo-EM) is not. Particularly worrisome is the very loose placement of ECL loops into the density, which is clearly noise from the micelle. Since this might be the first deposited FZD3 structure, it would be prudent to exercise caution and conservation with modelling and omit the regions that are not observed (more comments below).

The most exciting finding was that the forceful coupling of FZD3 to LRP5/6 through Wnt surrogates

promotes β -catenin signalling in the receptors that are not known for it.

While there are some parts of the manuscript that are quite interesting, many aspects are rushed to conclusion at the expense of the experimental evidence. In my opinion, this manuscript needs to be reworked to omit the parts in question. Unfortunately, without them the story becomes less exciting.

Response: We thank the reviewer for the overview and constructive comments/suggestions. We agree that the extracellular region of FZD3 is in poor electron density due to its well-known flexibility. We have now rephrased the text (detailed in following section) and added experimental evidence showing that the FZD3 "linker" domain interacts with Wnt5a "finger-3", which further validated our structure suggested model of the possible FZD3 CRD orientation.

We have further refined the structure model, removed residues with poor map densities in the flexible loop regions. We thank the reviewer for pointing out more references showing FZD antibodies can have potential for cancer therapy, we have added a sentence in the introduction, and updated the references (highlighted in the revised text). It reads: "Wnt receptor antibodies and other synthetic Wnt receptor-binding proteins are being developed with the intention of suppressing Wnt signalling by competing with Wnt for receptor binding, with the potential for tumour suppression (ref#29-31)."

We agree there are several other FZD CRD binding antibodies that have been developed, but our intracellular Nb9 is the only known intracellular binding molecule for FZDs.

Major issues:

1.Placement of the CRD domain into the density. A fairly significant portion of the manuscript is dedicated to identifying the CRD region in the cryo-EM map of FZD3-Nb9 and the implications of such CRD placement for Wnt binding. However, I question the validity of this structure (see also the following comment). While there is no doubt that authors have a CRD domain somewhere in that overall region in their cryo-EM map, it is still incredibly flexible. The density is very fuzzy and does not correlate to the determined FZD3 CRD structure. While the overall region appears correct, there is no way to orient the CRD pdb in any particular way to fit the density with confidence. The fit would be equally good (and wrong at the same time) if it was rotated 180° or upside down. The current model does not have a good model-map correlation. Most likely, the observed "fuzz" of the CRD domain represents an amalgamation of many unresolved conformations (or possibly even noise- see below), making any predictions about its orientation relative to the TM unfounded. This information then also invalidates any predictions about Wnt binding at the full-length FZD receptor (Fig. 3c) and any discussion about the role of the linker in Wnt binding and recognition. This entire section "The FZD3 extracellular domain and Wnt specificity" is not supported by experimental data.

Response: We thank the reviewer for the critical comments. It is well known that the FZDs CRD are naturally "incredibly flexible", as their "linker" sequences are significantly longer than that of smoothened (the only F-class GPCR family member with structurally visible CRD relative to its TMD). So far, little information is available for FZDs, although there is a proposed model based on fuzzy 2D classification of FZD5 (Reference#12). However, this long sought after information is important for understanding mechanisms of Wnt signalling, so we are trying our best to suggest a possible model based on observed extracellular density. We agree with the reviewer that our CRD map is poor density for modelling structure in precision. On the other hand, as noted by the reviewer "no doubt that authors have a CRD domain somewhere in that overall region in their cryo-EM map". We take the reviewer's point and rephrase corresponding content in the abstract and main text. In abstract (page2) it reads: "The cryo-EM structure of FZD3 in complex with Nb9 reveals a partially resolved density for the CRD, which exhibits positional flexibility, and a transmembrane conformation that resembles active GPCRs." and in result section (page 9-10), it reads "Although the weak extracellular density (even after the improvement with 3DFlex reconstruction ref#42) is not sufficient for confidently modelling the FZD3 CRD structure, the presence of CRD in this region is a certainty. Since Wnt and the Nb8 (raised against full length FZD3 antigen) should bind without sterically clashing with the TMD, we proposed a CRD orientation (Fig.3b), whereby our crystal structure can be most likely placed into the density."

And, we added experimental data to validate our structural suggested CRD orientation (for which the FZD3 linker is close to Wnt finger-3). Now (page 10) it reads: "To gain experimental evidence, we synthesized the biotinylated Wnt5a finger3 peptide and FZD3 linker peptide (Supplementary Fig.8a, d). The BLI data shows dose dependent responses of interaction with affinity $K_d:31.0 \pm 4.1 \mu\text{M}$ (Fig.3f), while the BSA control shows no or negligible responses with the same concentrations (Fig.3g). This validates our proposed model for the FZD3 CRD orientation. For the first time, it shows a Wnt finger3 interacting with a FZD through a linker domain and sheds light on

how FZD linker domain may contribute to Wnt signalling selection ref#7”.

Our proposed CRD orientation is about 90 degrees different from the proposed FZD5 CRD orientation. To turn our CRD 90°clockwise, the Nb8 (nanobody generated from full length FZD3) would clash with the TMD. To turn 90° anticlockwise, the Wnt would clash with the TMD. We agree with the reviewer that we cannot exclude the possibility of CRD rotating 180°, however, we have now got the additional experimental evidence that the FZD3 linker domain does interact with the Wnt5a “finger-3”, thus our current placed orientation is the most likely.

2. The quality of the cryo-EM data is quite problematic. The FSC curves (Fig. 6) show an incomplete drop to 0 for both 3D Flex and the Local refinement. The latter is particularly worrisome as the FSC has an additional shoulder that brings the resolutions to 2.94Å, far above what one expects from the curves with that drop to 0 as expected. This shoulder is most likely due to the correlation of the noise at higher frequencies or a subset of bad particles present in the dataset. However, the noise explanation is more likely considering the quality of the maps, particularly in the ECD regions. The map does not correlate to the model, and most of the loops are missing from the EM density despite being present in the model. The incompleteness and the lack of density for the loops is quite severe, and even the backbone is untraceable in most of ECL1, ECL2 and ECL3. Notably, the conformation of ECL1 and ECL3 is completely different even between the two structures authors used themselves- the one from consensus refinement (and presumably deposited in the PDB) and the one used to place CRD from the 3DFlex refinement.

As an extension of this issue, I wonder whether the observed CRD density is also due to noise correlation, considering that the FSC does not drop to 0 for that structure as well and the fact that the backbone of the CRD is untraceable and does not resemble even loose correlation to the map.

Response: We thank the reviewer for the careful examination of the structures. The observed 3DFlex reconstruction FSC curves showing incomplete drop to 0 is due to a “data reduction” step with Fourier data cropped from 512pxl to 300 pxl (as limited by our computing power at that time), which resulting in the full scale of the X axis was not being reached. We have now been able to use non-cropped data for the 3DFlex reconstruction (with updated computing capability) and now the curve can be dropped to 0, although the overall map quality remains similar. We have updated the curve in the supplementary Fig.8. The local refinement curve seemed satisfactory to us. We believe that the FSC “additional shoulder” is likely caused by detergent disc density signals in phases randomization. We have now supplied the original “Non Uniform refinement” FSC curve. The extracellular CRD and loops are indeed flexible, following the reviewer’s advice, we have removed the loop flexible residues with poor densities and refined the model (which will be shared with the editor and passed to reviewers for examination). The CRD fitting is explained above and the modified text makes it clear that the CRD placement in the density is not a precise structural model, just the most likely orientation when considering all other possibilities and the newly added experimental data. We modified Fig.3b to reflect the CRD flexibility. We agree with the reviewer, the fuzzy CRD from the 3DFlex refinement is not good enough to model the CRD structure and additional connecting loops, thus our structural model does not include these regions. We further refined the ECL1,2,3, including the part of the linker domain we used for our experiment, and the model will be shared with the reviewer for re-examination.

3. The BLI experiments showing the competition of Nb8 and Wnt5a are very confusing (Fig. 1d). Why did authors need a dissociation step following Nb8 binding to FZD3 CRD if they intend to look at the competition between Wnt5a and Nb8? Why is the dissociation of Nb 8 not complete here, even though it appears to come off completely in Fig.1a? The difference between Wnt5a binding to Nb8 bound and unbound FZD3 CRD does not look like much. How did authors determine that “the Nb8-dipped tip showed reduced binding to Wnt5a, indicating that Nb8 and Wnt5a compete for FZD3 binding”?

Response: We thank the reviewer, indeed, including the dissociation step was a bad idea. We have now performed the BLI assay without the dissociation step and updated Fig.1d. The competition is now much more pronounced.

Minor issues:

1. Representation of the 3DFlex map colored by local resolution (figure 6 middle right). Currently the range for the coloring is 2.5-3.5 Å, which misleadingly represents the entire CRD domain as having resolution at 3.5Å. This range needs to be extended significantly lower to showcase the true resolution of that region (perhaps to the range of calculated during the local resolution job).

Response: We thank the reviewer for this point. We have now included the original non-uniform

refinement map, TMD-Nanobody local refined map, and the 3D Flex map with CRD close up in a resolution range from 3 to 10 Å. (Supplementary Fig.6 updated).

2. What is the source of Wnt5a? and how was it prepared?

Response: Wnt5a is very difficult to purify in our hands. We purchased it from R & D systems. We have now made this clear, in page 22, the “Biolayer interferometry” section of the methods. It reads: “Wnt5a (purchased from Biotechne R&D systems, cat# 645-WN-010/CF)”.

3. Add labels to the concentrations on Fig.1

Response: We have now added concentration labels in Fig.1d. All the other panels concentration series were labelled in corresponding colour lines with numbers in nM.

4. Some ligands in the x-ray model have hydrogens (e.g. glycerol). Please be consistent.

Response: We have now removed the hydrogens and updated the model in the submitted PDB.

5. Please state how was FZD3 prepared for immunization (buffer? Detergent? Nanodiscs? Etc.).

Response: We have added the details in the method “Discovery of the nanobodies” section (page 18-19), now it reads: “Briefly, one llama (Lama glama) was immunized with purified recombinant FZD3 in 10 mM HEPES pH7.5, 150 mM NaCl, 1% glycerol, 0.03% DDM, 0.006% CHS. All selections were done in the same buffer.”

6. How was cryo-EM map sharpening done?

Response: We did the map sharpening in cryosparc with DeepEMhancer with wide target model. This is stated in the supplementary Fig.6. and supplementary Table 2.

7. Describe how masks were generated for 3DFlex and local refinement.

Response: For the 3D Flex reconstruction, we firstly used Chimera “Segger” to define segmentations of FZD3_megabody over the Gaussian filtered volume. We defined CRD, TMD/disc, Nb9 and megabody sections. The segmentation file was then imported into cryosparc 3D Mesh preparation program with the segment connections defined. The segment mask looked like the following:

[Redacted]

And the segment mesh looked like the following:

[Redacted]

For the local refinement, we also generated mask with Chimera “segger”, excluded the Megabody scaffold and the CRD part, leaving only the TM and nanobody densities, then filtered with Gaussian width 2.5. The volume was imported into cryosparc and converted to a mask with the volume tool.

8. There is a weird density (or abundance of it) near the intracellular parts of TM5 and 6.

Please comment on this.

Response: We cannot not be sure about what they are exactly. Given that some high resolution membrane proteins show lipids in their transmembrane region, this density may be contributed by lipids, however, in our map the resolution is not sufficient to fit any lipids or cholesterol.

Reviewer #3 (Remarks to the Author):

The authors present an in-depth structural analysis of two nanobodies targeting the FZD3 receptor using multiple structural techniques, crystallography and cryo-EM. The binding interfaces of both an intracellular and extracellular nanobody were resolved with residue level precision, which will facilitate further tool development. Nb8 binds to the CRD region of FZD3, while Nb9 binds to the putative DVL and G protein binding site. The engineered Nb8 (Nb8-DKK1C) is able to induce activation of the canonical signaling pathway, while FDZ3 is known to activate exclusively to non-canonical signaling pathways.

Although Nb8, the engineered Nb8 and Nb9 are interesting tools, the structures, biophysical and cellular assays presented in this manuscript do not, as suggested, provide insight into the

mechanisms of FZD3 signaling. Moreover, data on the mode of action of the nanobodies as modulators on basal and Wnt5a-induced signaling in the non-canonical and canonical signaling assays is limited or lacking.

Response: We thank the reviewer for the overview. With regard to new insights into the mechanisms of FZD signalling, our additional experimental data shows that a Wnt5a “third finger” (beyond the two other fingers well known for FZD interaction) can also interact with FZD3 through the linker domain. This observation sheds new light on the previously unappreciated role of Wnt finger-3 in FZD recognition. It may help to solve the mystery about how FZD linker domain could contribute to Wnt selection.

1. The current data show that Nb8 binds to the site Wnt5a is binding to as it impairs binding of Wnt5a to FZD3 as shown in the BLI studies. FZD3 is reported to signal exclusively through non-canonical Wnt pathways, without eliciting β -catenin-dependent signaling. Does Nb8 inhibit the functional responses induced by Wnt5a in non-canonical signaling assays?

Response: We thank the reviewer for this intriguing question. Wnt canonical signaling can be measured quantitatively in well-established TOP flash assays, however there are no equivalent reliable assays available for non-canonical signaling. The existing ATF2-luciferase assay for non-canonical Wnt signaling seems to only work well in vivo in model organisms (such as frog, zebrafish), but is not able to give reliable results in cultured mammalian cells. We tried several possible non-canonical assays, with thorough controls (thanks to the recent availability of FZD1-10 knock-out HEK293 cells). We observed some Wnt5a induced non-canonical assay signals that may not be dependent on FZDs. So, we do not have experimental evidence to show if Nb8 can perturb non-canonical signaling by competing with Wnt5a. However, given our structural and BLI data indicating that Nb8 can compete with Wnt5a, it is reasonable to speculate that the Nb8 can inhibit the non-canonical signaling mediated by FZD3, but this awaits direct confirmation.

2. The engineered Nb8-DKK1C nanobody appears to induce a canonical response, activating TOP flash reporter activity. Although this receptor has reported not to signal to this pathway include Wnt5a as a (negative) control. What effect does this engineered Nb8 elicit in the non-canonical β -catenin assay(s)?

Response: We agree, it is well established that FZD3/6 receptors do not trigger TOP flash activity by Wnt3a or any other Wnts. It is believed that bridging FZD with co-receptor LRP5/6 is required for canonical signaling (TOP flash activity). Canonical Wnts (such as Wnt1,3/3a, 8) may not be able to bind FZD3/6 well, while the non-canonical Wnts (such as Wnt5a) may not be able to bind LRP5/6 well. Our surrogate has DKK1c to bind LRP6 and Nb8 to bind FZD3, and results in TOP flash activity. This suggests that FZD3/6 themselves can generate canonical activity as long as they can be brought into proximity with LRP5/6. Similarly, an anti-FZD6 antibody based surrogate (a FZD6 and LRP6 multivalent antibodies based surrogate) has been demonstrated to be as good as canonical FZD based surrogates in the case of lung stem cell regeneration (ref#38: Cell, 2023, 186(14):2995-3012).

As explained above, we do not have a reliable non-canonical assay to test if the engineered Nb8 could elicit non-canonical signaling. Given LRP5/6 may not be involved in non-canonical signaling, (ROR1/2, RYK, PTK7 may be involved instead), we assume our Nb8-DKK1C cannot elicit non-canonical signaling.

3. It would be informative to provide a more clear illustration (Fig.2e) of the Nb8-DKK1C nanobody fusion protein.

Response: We thank the reviewer for the good suggestion. We have increased the size of the diagram and now placed it in a separate panel, Fig.2e to make it clearer.

4. The structural superposition of FZD3 CRD Nb8 complex onto the FZD8 CRD dimer indicates that Nb8 binding would hinder formation of the CRD dimerization interface (Fig. 2d). Is there any evidence FZD3 CDR exists as dimer or dimerization is important?

Response: We thank the reviewer for raising this point. Yes, FZD CRD dimerization is likely to be important for signalling. We have added comments at the end of the “Crystal structure of FZD3 CRD in complex with Nb8” section (from the end of page 7), it now reads: “FZD CRD dimerization induced by Wnt palmitoleic lipid has been documented^{10, 33, 34}, and its importance in Wnt signalling has been further demonstrated by multivalent Wnt surrogates³⁵.” And at the discussion section (Page 16), also a paragraph discusses this point.

FZD CRD dimerization is usually observed when complexed with Wnt (lipid). Some FZDs, like FZD4 CRD protein may pick up lipids from cell culture media and crystallize as a dimer. In our FZD3 CRD protein purification as judged from the gel filtration profile, it is in a monomeric state.

5. When FZD3 is co-transfected with DEP-mVenus, fluorescence is mainly restricted to the plasma membrane, indicating basal association of the DEP-mVenus (Fig. 4i). Can this be modulated by Wnt

ligands, Nb8 or Nb8-DKK1C?

Response: We thank the reviewer for the interesting question. Dishevelled (DVL)'s DEP domain binding to FZD is essential for Wnt signalling. There is a basal association of DEP to FZD, and it is known that Wnt ligands promote DEP domain swapping, and swapped DEP masked the FZD binding epitope and may disassociate from FZD ref#48 and reference (not cited in the paper due to the number limitation): Beitia GJ et al. 2021, Regulation of Dishevelled DEP domain swapping by conserved phosphorylation sites, PNAS 118(26):e2103258118). Nb8-DKK1C is basically a Wnt mimic, thus likely it would behave similarly to Wnt. In our report, we focused on Nb9 which was observed binding at the proposed DEP binding interface from intracellular side. How mechanistically extracellular binders (like Wnt, Nb8 or Nb8-DKK1C) trigger the intracellular DEP binding conformation change is a big future challenge for the Wnt field.

6. FZD3 was previously reported to be able to signal through G α s. In Fig. 5C, a decrease in the BRET ratio between G and G is expected for G protein activation. Therefore, the results shown seem to suggest that FZD3 interferes with basal G protein activation or coupling and instead promotes reassociation when unobstructed. I would not call this coupling in a traditional GPCR sense. Signaling by Wnt5a should be taken along as a control.

Response: We thank the reviewer for this point. We totally agree that our observed lower BRET value indicates that the Nb9 may interfere with the basal G proteins attaching to the receptor, which is different from the lower BRET value resulting from traditional GPCR activation where ligand induces coupled G protein subunits to dissociate. We have added this point at the end of the section of "Nb9 interferes with heterotrimeric G proteins coupling to FZD3" (page 15). It reads: "Of note, this lower BRET value is in contrast to traditional GPCR activation BRET assays in which a lower signal can result from ligand binding inducing coupled G protein subunits to dissociate."

7. Why is the structure of FZD3 superimposed in Fig 3d onto the active and inactive structure of SMO and not the constitutively active FZD7-mGs, and inactive structures of FZD4 and FZD5?

Response: We thank the reviewer for the suggestion, we have now superimposed with active FZD7 and inactive FZD5 structures, and updated Fig.3d.

8. On pg. 10, RMSD is presented twice without indication of what atoms are being compared. Additionally, the authors indicate that there is only a 0.2Å difference in RMSD when comparing their complex to active or inactive structures. From this analysis, it seems difficult to conclude anything regarding the similarity of the presented complex to previous structures as is claimed in the following line. Comment on why the RMSD analysis gives such similar results for both conformational comparisons or alter the analysis appropriately.

Response: We thank the reviewer for this point. Indeed, the overall structure superimposition may not reflect the local movement. We have modified the text to read: "For example, at the cytoplasmic end of TM6, FZD3 K4136.25 has a C α distance of 6.4 Å to FZD5 K4426.25 but only 1.6 Å to FZD7 K4636.25, (superscript numbers refer to the Ballesteros and Weinstein numbering system), despite the overall structures being similar among the three (FZD3 shows an RMSD of 1.6 Å and 1.8Å to active FZD7 and inactive FZD5 respectively, when superposed with all the 332 residues of TMD helices and loops)."

9. Nanobodies were selected using phage display. Interestingly, this resulted in 5 out of 7 lead molecules with almost identical sequences. It would be interesting to the field to provide more information on how these 7 were selected, amount of receptor used for panning and on which basis the 7 molecules were selected. How many rounds of panning, clones picked and how often did the individual sequences appear amongst these? This might reveal information about a potential bias towards specific epitopes on FZD3.

Response: We thank the reviewer for this suggestion. We have now added more details in Method, section of "Discovery of the nanobodies", page 18-19. It reads: "Briefly, one llama (*Lama glama*) was immunized with purified recombinant FZD3 in 10 mM HEPES pH7.5, 150 mM NaCl, 1% glycerol, 0.03% DDM, 0.006% CHS. All selections were done in the same buffer. Panning was performed on 1 μ g of FZD3 full or 1 μ g of FZD3 TMD coated on maxisorb wells or on neutravidin-captured, biotinylated FZD3 (400nM). FZD3 specific phages were recovered by incubating the coated wells with Trypsin (250 μ g/ml) for 30min. Trypsin was neutralized by adding AEBSF and phages were added to freshly grown TG1 cells. After 2 rounds of selection, individual colonies from all selection conditions were screened for the expression of FZD3 specific nanobodies in an ELISA. For the detection of the presence of specific Nanobodies, the "EPEA"-tag at the end of the Nanobody was used (LifeTechnologies #7103252100) in combination with Streptavidin Alkaline Phosphatase (Promega #V5591). Positives were detected by using 4-Nitrophenyl phosphate disodium salt hexahydrate (2mg/ml). The absorption at 405 nm was measured 60 min after adding the enzyme substrate p-nitrophenyl phosphate. 171 clones were sequence analysed to reveal enriched families. Family 1 (Nb1,3-6) was identified 74 times, family 2 (Nb8) 39 times and family 3 (Nb9) 5 times. Family 3 was identified as a binder to the TMD."

10. For the cross reactivity check on the different FZD CRDs, the authors nicely checked all 10 CRDs.

Based on Suppl Fig 4 and the binding data, can the authors deduct a potential epitope from the cross-reactivity with FZD6

Response: Thanks for raising this point, yes, we can deduct a potential epitope from the cross-reactivity with FZD6. It has now shown in the Supplementary Fig. 5 figure and the figure legend states: "The FZD3 amino acids N70, K112 are uniquely shared the FZD6, while the D72, E109, and E115 are substituted with similar residues in the FZD6, these positions are potential epitopes for cross-reactivity of Nb8 with the FZD6."

11. Binding of Nb1 to full length FZD3 was tested but affinities were not calculated. Given that this is the largest cluster of sequences in Suppl Fig 1, it might be assumed that Nbs1-7 actually have the highest binding affinity towards FZD3. Can the authors elaborate on how these relatively low signals in BLI (less epitopes available) and the relatively high abundance in panning outputs?

Response: We thank the reviewer for this intriguing immunology question. We believe that abundance in panning is not necessarily directly associated with affinity. The overall Nb1 BLI response to full length FZD3 is very low (supplementary Fig. 3). Although even this low response can in principle be used to calculate affinities, we are concerned that values might be misleading and so leave it uncalculated. Actually, before we got the Nb9-FZD3 structure, we were looking at the Nb1_megabody_FZD3 grids, and found that FZD3 (visible as detergent discs) does not come bound with the Nb1_megabody, consistent with low affinity. We therefore focused on the Nb9_megabody.

12. In line 136, the authors indicate the contact surface between Nb8 and FZD3 CRD. How do these correlate to those from other Nb-antigen interfaces? And can the authors elaborate on the nature of the interactions in this surface and how this correlates to the relatively high affinity? Like the Methionine?

Response: We thank the reviewer for the interesting question. The contact surface area is only one of several parameters contributing to affinity and we agree with the reviewer, the interaction nature, such as hydrophobicity, may contribute more to affinity and should have been mentioned. After the stated surface area sentence, we have therefore added: "The interactions are predominantly hydrophobic (Supplementary Fig. 5). For example, FZD3 M116 form hydrophobic interaction with Nb8 Y37, N96 and L98. Hydrogen bonds also form between FZD3 K112 with Nb8 E103 and D105."

13. Line 145. Can the authors specify the interactions partners of M116 and K112?

Response: In Supplementary Fig. 5b, we have highlighted the interaction of FZD3 M116 and K112 with Nb8. M116 is unique to FZD3, while K112 is shared with FZD6. In the figure, we colored Nb8 CDR1 in blue, CDR2 in green and CDR3 in red (to be consistent with Fig.2). The binding residues for M116 are Nb8 CDR1 (Y37), CDR2 (A50), CDR3 (N96, L98). The binding residues for K112 are Nb8 CDR3 (E103, D105, L98). We have added details in the Supplementary Fig.5 legend.

14. Line 193. Limited information is available on how the megabody was designed. In between which residues was the YgjK inserted and how was this construct characterized. Did this insertion affect the binding affinity Nb9?

Response: We thank the reviewer for raising this point. We have now added more details at the end of "Discovery of the nanobodies" section of the methods (page 20), and referred to the original publication for more details, Ref#40). The added text reads: "Briefly, the nanobody was PCR amplified with the primers: TU89: 5'-CCTTGAGCTCTTCGTCCTGAGACTCTCCTG-3' and EP230: 5'-AGGACTGCTCTTCCACTGGAGACGGTG ACCTGGGT-3'. The PCR product was digested with SapI and ligated with SapI/SalI double cut vector. Transformed WK6 bacteria were used for megabody production after a sequencing check. The resulting expressed protein contains (from the N-terminus) the first β -strand of the nanobody (AA 1-13, an universal sequence for all nanobodies embedded in the vector), YgjK (AA24-783, with a flexible linker in the middle), and ended with the nanobody sequence starting SLRLSC (i.e. omitting the first β -sheet)."

Structurally, the fusion scaffold is on the opposite side of a nanobody's CDRs, thus minimizing the possibility of affecting antigen binding. And from published evidence, the fusion didn't affect the binding. (example reference #41). We have added to the description in the main results section of "Cryo-EM structure of FZD3 in complex with Nb9 megabody" and a reference citation in page 9. It reads: "The scaffold is fused on the opposite side of the nanobody to its CDRs, thus minimizing the possibility of affecting antigen binding. For a similarly fused megabody (with a nanobody targeting Hedgehog acyltransferase) the fusion has been demonstrated to have little effect on binding affinity (ref#41)"

Additional points

15. The binding affinities (lines 109 and 111, 121-124, lines 282 and 283) should be indicated with SD or SEM and N.

Response: We have updated all the affinities numbers in full. These are BLI data analyzed results, not traditional statistical data, so no N number is given.

16. The superimposition of the CRDs mentioned on pg 6 should be shown somewhere or the figure needs to be called.

Response: We have now called out Supplementary Fig.4 on Page 6, thank you.

17. Line 162, CDR dimerization should be CRD dimerization.

Response: We have corrected it, thank you.

18. Pg. 19, GDN is not defined in text.

Response: Thank you, we have now defined GDN, glyco-diosgenin, in the text (page 21).

19. In Fig. 1, the binding curves in e and f are poorly fit and perhaps the model used needs to be reconsidered.

Response: We agree the fit is poor, these two are full membrane proteins loaded on tips, we are not sure why the sensor-gram traces were poor. Given that our structure showed the binding to be 1:1, we used a conservative approach to fitting, i.e. a simple 1:1 model.

20. Fig. 4f, the binding curves are clearly biphasic and poorly fit with the 1:1 model used.

Response: We agree the fit is poor. As above, it is a membrane protein on the tips, consistently showing poor behavior. Again, we thought it best to use a simple 1:1 model.

21. Fig. 5, the title states "... a Nb8-based Wnt surrogate activates canonical signaling", however this data was not presented in the figure or elsewhere.

Response: We thank the reviewer for spotting this mistake, which was left after the figure was moved to Fig.2e.f. We have removed it.

22. Supp. Fig. 3: It is unclear what the numbers in the legend are indicating and panel b has no legend. Additionally, the Nb9 traces in panel b are all the same shade of gray and indistinguishable.

Response: We have changed Supp. Fig.3 b, coloured Nb9 traces in panel b, and updated the figure legend.

23. Supp. Fig. 5: The magenta used for Nb8 in panel b should be changed so as to make the red oxygen atoms easier to see.

Response: We thank the reviewer for the suggestion and changed Supp.Fig.5b Nb8 sticks to light pink, now the oxygen atoms are easier to see. The figure has been updated.

24. The authors inconsistently use British and American spellings, sometimes within the same sentence. This needs to be addressed.

Response: Thank you, we have now used a British English spelling check to correct them.

Version 1:

Reviewer comments:

Reviewer #1

(Remarks to the Author)

the authors have addressed all my comments satisfactorily

Reviewer #2

(Remarks to the Author)

All of my comments have been addressed. I thank the authors for taking on board the critique regarding the uncertainty of CRD placement and performing additional experiments to confirm the pose. This has strengthened the manuscript and provided much-needed confidence in the structural data. This is a good quality work, and I am looking forward to seeing it published.

Reviewer #3

(Remarks to the Author)

No additional experiments of newly identified nanobodies on non canonical signaling and FZD3 and DEP-mVenus localisation and insight on the biology of these receptors have been included in the revised manuscript. Noncanonical signaling and impacts of the nanobodies on FZD3 biology would significantly enhance the impact of this manuscript. The interaction between WNT5a_finger3 and the linker domain has been put forward as the main structural discovery. Is the interaction between the WNT5a_finger3 with the linker domain relevant to activation? Is there evidence that this interaction occurs with the full receptor or only with peptides?

- Fig 5 'the Nb8-based Wnt surrogate activates canonical signaling' is still included in the legend of fig 5.

The G protein response is still not adequately explained. Does FZD3 inhibit G protein activation by other receptors? The BRET experiments suggests that the receptor suppresses Galpha beta/gamma dissociation, while the nanobody promotes G protein activation. Therefore, it is not clear how the binding of the nanobody interferes with G protein coupling. Zhang 2024 suggests that FZD3 shows constitutive G protein activity using the same assay, how do you reconcile your results with theirs?

- Evidence of monomeric state of FZD3_CRD should be presented. The rebuttal claims gel filtration results are available, these should be added.

- Citation limit is not a reasonable excuse to not cite relevant papers, please include all appropriate references.

- It is still unclear what atoms are being compared in the RMSD analysis on pg. 11.

Author Rebuttal letter:

Reviewer #1 (Remarks to the Author):

the authors have addressed all my comments satisfactorily

Reviewer #2 (Remarks to the Author):

All of my comments have been addressed. I thank the authors for taking on board the critique regarding the uncertainty of CRD placement and performing additional experiments to confirm the pose. This has strengthened the manuscript and provided much-needed confidence in the structural data. This is a good quality work, and I am looking forward to seeing it published.

Reviewer #3 (Remarks to the Author):

No additional experiments of newly identified nanobodies on non canonical signaling and FZD3 and DEP-mVenus localisation and insight on the biology of these receptors have been included in the revised manuscript. Noncanonical signaling and impacts of the nanobodies on FZD3 biology would significantly enhance the impact of this manuscript.

The interaction between WNT5a_finger3 and the linker domain has been put forward as the main structural discovery. Is the interaction between the WNT5a_finger3 with the linker domain relevant to activation? Is there evidence that this interaction occurs with the full receptor or only with peptides?

Response: We completely agree with the reviewer that gaining further insights into FZD3 nanobodies in noncanonical signalling would significantly enhance our understanding of FZD3 signalling. Due to the current lack of reliable, sensitive assays for noncanonical signalling, we anticipate that the nanobodies reported here will serve as excellent tools for defining FZD3 activation mechanisms once reliable, sensitive assays are developed in the future.

We believe that the interaction between WNT5a_finger3 and the FZD linker domain contributes to Wnt activation. It is documented that the FZD linker domain plays a role in Wnt recognition and selection (reference #7), although the specific

part of Wnt responsible for this interaction had not been identified until this report. It is well known that Wnts use two fingers (thumb and index) for FZD interaction. Manipulating these two sites would likely disrupt the correct folding of the Wnt protein. Therefore, assessing the contribution in Wnt signalling of the third finger without the other two is challenging. However, reference #7 provides clear evidence that the linker domain contributes to Wnt recognition specificity and canonical-noncanonical switching. By using peptide interactions, we obtained direct, clean results and solved the mystery of which part of Wnt is responsible for the linker domain interaction.

- Fig 5 'the Nb8-based Wnt surrogate activates canonical signaling' is still included in the legend of fig 5.

Response: Thank you to the reviewer, and our apologies for the oversight. We have now removed the remaining text from the figure legend title.

The G protein response is still not adequately explained. Does FZD3 inhibit G protein activation by other receptors? The BRET experiments suggests that the receptor suppresses Galpha beta/gamma dissociation, while the nanobody promotes G protein activation. Therefore, it is not clear how the binding of the nanobody interferes with G protein coupling. Zhang 2024 suggests that FZD3 shows constitutive G protein activity using the same assay, how do you reconcile your results with theirs?

Response: We thank the reviewer for raising this issue. As Wnts may not (or have not been shown to) trigger G-protein signalling, it is possible that other unidentified agonists may serve as G-protein signalling triggers. Indeed, Zhang et al. (2024) (reference #54) used the same assay, but their net BRET value is negative, while ours is positive. We have consulted the BRET2 assay inventors (Nicholas Kapolka nkapolka@email.unc.edu and Bryan L. Roth bryan_roth@med.unc.edu) and confirmed that our observations and explanations are correct: a positive (higher) value indicates G-protein association, while a negative (lower) value indicates disassociation. We believe Zhang et al. (2024) made an error, perhaps confused by the lower values in traditional GPCR signalling. In traditional GPCR signalling, initially associated G-proteins disassociate (lower net BRET value) upon agonist engagement, resulting in GDP/GTP exchange (GPCR activation). The nanobody (Nb9) prevents the initial G-protein association step, leading to negative (lower) values in the assay. However, this constitutive negative value does not involve GDP/GTP exchange (activation). We have added this explanation at the end of the paragraph on the bottom of page 15. It reads: "Notably, this lower BRET value contrasts with traditional GPCR activation BRET assays, where a lower signal can result from agonist-induced disassociation from the coupled G protein subunits with GDP/GTP exchange events. Nb9 prevents the initial association step and thus may disrupt agonist-induced GDP/GTP exchange required for GPCR activation."

- Evidence of monomeric state of FZD3_CRD should be presented. The rebuttal claims gel filtration results are available, these should be added.

Response: Thanks, we have now added the gel filtration profile in the Supplementary Fig.4d, and the main text called the figure at the top of page8. It reads: "Our FZD3 CRD alone appears to be monomeric as judged from the gel filtration profile (Supplementary Fig.4d). However, this does not exclude it potentially forming a dimer upon ligand engagement similar to the behaviour of FZD4 CRD, which itself is monomeric⁸, but forms a dimer in response to Wnt5a binding."

- Citation limit is not a reasonable excuse to not cite relevant papers, please include all appropriate references.

Response: We have added more references, bringing the total to 70, including the new publication from Zhang (2024), which was published while our manuscript was in the revision stage. We are doing our best to include the most relevant references fairly. As our manuscript covers many aspects of Wnt signalling related to crystal and cryoEM structures, functional assays (BLI, luciferase, nanobodies, surrogates, DVL assays, and G protein BRET assays), the number of references could easily exceed a thousand. We apologize for not being able to include all, but we are

happy to add more if the reviewer can narrow it down to a particular aspect.

- It is still unclear what atoms are being compared in the RMSD analysis on pg. 11.

Response: We used C α atoms of the 332 residues of the TMD helices and loops from FZD3, FZD7, and FZD5 for comparison. The revised sentence in page 11/12 now reads: "FZD3 shows an RMSD of 1.6 Å and 1.8 Å to active FZD7 and inactive FZD5, respectively, when superposed with all the C α atoms of the 332 residues from the TMD helices and loops".
